# Functional synergy of a human-specific and an ape-specific metabolic regulator in human neocortex development

Lei Xing [1,2] ✉, Vasiliki Gkini[3], Anni I. Nieminen [4], Hui-Chao Zhou[5], Matilde Aquilino[3], Ronald Naumann [1], Katrin Reppe[1], Kohichi Tanaka[6], Peter Carmeliet [7,8,9], Oskari Heikinheimo[10], Svante Pääbo [11,12], Wieland B. Huttner [1] ✉ & Takashi Namba [3] ✉

Metabolism has recently emerged as a major target of genes implicated in the evolutionary expansion of human neocortex. One such gene is the human-specific gene *ARHGAP11B*. During human neocortex development, ARHGAP11B increases the abundance of basal radial glia, key progenitors for neocortex expansion, by stimulating glutaminolysis (glutamine-to-glutamate-to-alpha-ketoglutarate) in mitochondria. Here we show that the ape-specific protein GLUD2 (glutamate dehydrogenase 2), which also operates in mitochondria and converts glutamate-to-αKG, enhances ARHGAP11B's ability to increase basal radial glia abundance. ARHGAP11B + GLUD2 double-transgenic bRG show increased production of aspartate, a metabolite essential for cell proliferation, from glutamate via alpha-ketoglutarate and the TCA cycle. Hence, during human evolution, a human-specific gene exploited the existence of another gene that emerged during ape evolution, to increase, via concerted changes in metabolism, progenitor abundance and neocortex size.

The human body and organs have experienced massive changes in their structure and function during evolution. The evolutionary expansion of the human brain, especially of the neocortex, is one of the most prominent examples of such changes[1]. Among primates, the human neocortex exhibits the biggest volume and largest number of neurons[2,3], which is considered to be a basis of human cognitive abilities[1,4]. Recent studies have implicated a number of human-specific and hominid-specific genes and mutations in the evolution of human brain size and brain morphology[5–12].

An increase in the abundance of neural stem/progenitor cells (NPCs) is thought to be a crucial developmental event to produce an increase in the number of neocortical neurons, hence contributing to the evolutionary enlargement of the human neocortex[13–17]. There are two classes of NPCs: apical progenitors (APs), mainly consisting of apical (or ventricular) radial glia (aRG), and basal progenitors (BPs)[16,18]. Of these two classes of NPCs, BPs are the main source of neuron production in the developing neocortex. There are two types of BPs: (i) basal intermediate progenitors (bIPs) and (ii) basal (or outer) radial

[1]Max Planck Institute of Molecular Cell Biology and Genetics, Dresden, Germany. [2]Department of Biological Sciences, University of Manitoba, Winnipeg, MB, Canada. [3]Neuroscience Center, HiLIFE – Helsinki Institute of Life Science, University of Helsinki, Helsinki, Finland. [4]FIMM Metabolomics Unit, Institute for Molecular Medicine Finland, University of Helsinki, Helsinki, Finland. [5]Center for Cancer Biology (CCB), VIB-KU Leuven, B-3000 Leuven, Belgium. [6]Laboratory of Molecular Neuroscience, Medical Research Institute, Tokyo Medical and Dental University, Tokyo, Japan. [7]Laboratory of Angiogenesis and Vascular Metabolism, Department of Oncology, KU Leuven, B-3000 Leuven, Belgium. [8]Laboratory of Angiogenesis and Vascular Metabolism, Center for Cancer Biology, VIB, B-3000 Leuven, Belgium. [9]Center for Biotechnology, Khalifa University of Science and Technology, Abu Dhabi, United Arab Emirates. [10]Department of Obstetrics and Gynecology, University of Helsinki and Helsinki University Hospital, Helsinki, Finland. [11]Max Planck Institute for Evolutionary Anthropology, Leipzig, Germany. [12]Human Evolutionary Genomics Unit, Okinawa Institute of Science and Technology, Okinawa, Onna-son, Japan. ✉e-mail: lei.xing@umanitoba.ca; huttner@mpi-cbg.de; takashi.namba@helsinki.fi

glia (bRG). In mammals with a small neocortex (e.g. mice), approximately 90% of the BPs are bIPs with relatively low proliferative capacity. In contrast, in primates, including human, half of the BPs are highly proliferative bRG. Thus, the increase in the relative abundance of bRG and in their proliferative capacity is thought to underlie the evolutionary expansion of the neocortex in human.

The human-specific gene *ARHGAP11B*, when ectopically expressed under the control of its own promoter in fetuses of the common marmoset *Callithrix jacchus*, a New World monkey, has previously been shown to increase (i) BP abundance, in particular bRG abundance, (ii) upper-layer neuron number, and (iii) the size of the fetal neocortex, and to induce its folding[19]. To address the relevance of these findings for the expansion of the neocortex during human evolution, in a recent study, ARHGAP11B expression and function were abolished in human cerebral organoids, and conversely this human-specific gene was expressed in chimpanzee cerebral organoids[20]. The findings of this study indicated that ARHGAP11B is essential and sufficient to ensure the high BP abundance that is characteristic of developing human neocortex[20]. These findings, in turn, have led to the conclusion that ARHGAP11B was a major, if not the main, contributor to the increase in BP abundance and hence the expansion of the neocortex that occurred during human evolution[20].

However, how comprehensive is the insight into ARHGAP11B action that has emerged from the above study that used human and chimpanzee cerebral organoids? In considering this question, it is important to realize that genes that evolved specifically in the hominid lineage are intrinsically present in chimpanzees and human cerebral organoid systems. Hence, two key questions arise: Is there a functional synergy between *ARHGAP11B* and another hominid-specific gene with regard to BP, and notably bRG, amplification that would promote ARHGAP11B action? And if so, what would be the mechanistic basis of such a synergy?

The ARHGAP11B protein has previously been shown to be imported into the mitochondrial matrix and to stimulate glutaminolysis, a metabolic pathway converting glutamine to glutamate to the TCA cycle intermediate metabolite alpha-ketoglutarate[21]. Both, the presence of ARHGAP11B in mitochondria and its stimulation of glutaminolysis, are essential for the BP amplification typical of humans[21], presumably reflecting the promotion of anabolic metabolism[22]. Here, we demonstrate that the ARHGAP11B-induced BP amplification is enhanced by GLUD2, an enzyme encoded by an ape-specific gene that catalyzes the conversion of glutamate to alpha-ketoglutarate. This constitutes an example of the functional synergy between a human-specific and an ape-specific gene, and provides further evidence for the emerging concept that changes in cell metabolism underlie the increase in the proliferative capacity of BPs and hence the expansion of the neocortex during human evolution[11,21–24].

## Results

### Human NPCs express the ape-specific protein GLUD2 that like ARHGAP11B operates in mitochondria

To explore a potential functional synergy between the human-specific gene *ARHGAP11B* and other, recently emerged genes in the increase in BP abundance during human evolution, we focused on metabolic regulators that operate in mitochondria. Among the previously identified 50 primate-specific genes preferentially expressed in human NPCs[8], we found only two genes with the GO-term "mitochondrial matrix" (GO:0005759) or "mitochondrion" (GO:0005739) (Fig. 1a). One was the human-specific gene *ARHGAP11B*[6,25], and the other was *GLUD2*, an ape-specific gene[26]. Interestingly, the protein encoded by *GLUD2* is glutamate dehydrogenase 2, a mitochondrial enzyme that catalyzes the conversion of glutamate to alpha-ketoglutarate (αKG)[26], which is the last step of glutaminolysis[27]. Like *ARHGAP11B*, *GLUD2* is more highly expressed

in both aRG and bRG than in neurons in the fetal human neocortex (Supplementary Fig. 1a, b)[28]. Therefore, we investigated whether there is a functional synergy of ARHGAP11B and GLUD2 in NPC abundance, notably BP and bRG abundance.

### GLUD2 enhances the ability of ARHGAP11B to amplify bRG, but not bIPs

To this end, we crossed two previously generated transgenic mouse lines, an ARHGAP11B-transgenic mouse line[29] and a GLUD2 BAC-transgenic mouse line[30], to generate ARHGAP11B + GLUD2 double-transgenic mice (heterozygous for each gene). ARHGAP11B + GLUD2 embryos express both ARHGAP11B and GLUD2 in the developing neocortex at E14.5, similar to the expression pattern observed in fetal human neocortex (Supplementary Fig. 1c–m). Immunofluorescence for the mitotic marker phospho-vimentin (pVim) revealed no changes in the number of pVim+ cells at the apical surface of the ventricular zone (VZ) of the lateral neocortex at E14.5 among the four genotypes studied (WT, ARHGAP11B, GLUD2, ARHGAP11B + GLUD2) (Fig. 1b, c). This suggested that neither ARHGAP11B nor GLUD2 nor both proteins together affect the abundance of mitotic aRG/APs. In contrast, the number of pVim+ abventricular cells without basal and/or apical processes, considered to be mitotic bIPs, was significantly increased in both ARHGAP11B and ARHGAP11B + GLUD2, but not GLUD2, embryos compared to WT embryos (Fig. 1b, d). This increase was of the same magnitude in ARHGAP11B and ARHGAP11B + GLUD2 embryos (Fig. 1b, d). In contrast, when the number of pVim+ abventricular cells with basal and/or apical processes, considered to be mitotic bRG, was quantified, we observed a significant increase in the ARHGAP11B + GLUD2 embryos not only compared to WT and GLUD2 embryos, but also compared to the ARHGAP11B embryos (Fig. 1b, e). When the number of mitotic bRG in ARHGAP11B embryos was compared only to that in WT embryos, a significant increase was observed (not indicated), as previously reported[29], whereas this was not the case when the number of mitotic bRG in GLUD2 embryos was compared to that in WT embryos. These data therefore suggested that GLUD2 enhances the ability of ARHGAP11B to increase the abundance of mitotic bRG, but does not enhance ARHGAP11B's ability to increase the abundance of mitotic bIPs.

Because the genomic GLUD2 locus as present in the GLUD2 BAC-transgenic mouse line contains a long noncoding RNA (lncRNA)[30], we next examined whether the increase in mitotic bRG abundance observed in the ARHGAP11B + GLUD2 embryos compared to the ARHGAP11B embryos (Fig. 1b, e) was due to the expression of GLUD2 or the expression of the lncRNA. To this end, mouse embryonic neocortex at E13.5 was *in utero* electroporated with expression plasmids containing the cDNAs encoding either the ARHGAP11B protein or the GLUD2 protein, together with a GFP-expressing plasmid, and subjected to immunofluorescence at E14.5. A significant increase in the percentage of GFP+ cells that (i) express the mitotic marker PH3, (ii) exhibit an abventricular localization, and (iii) possess basal and/or apical process (as revealed by GFP immunofluorescence), i.e. that are mitotic bRG, was observed upon ARHGAP11B + GLUD2 double-electroporation compared to control (empty plasmid), to ARHGAP11B, and to GLUD2 single-electroporation (Supplementary Fig. 2a, b). Again, when the number of mitotic bRG in ARHGAP11B single-electroporated embryos was compared to that in control embryos, a significant increase was observed, whereas this was not the case when the number of mitotic bRG in GLUD2 single-electroporated embryos was compared to that in control embryos. This indicated that the GLUD2 protein alone is able to enhance the ability of ARHGAP11B to increase the abundance of mitotic bRG.

To examine whether the increase in mitotic bRG was associated with an increase in bRG abundance, E14.5 mouse neocortex of the four different genotypes was analyzed by immunofluorescence for Pax6, a marker of APs and proliferative BPs including bRG in the

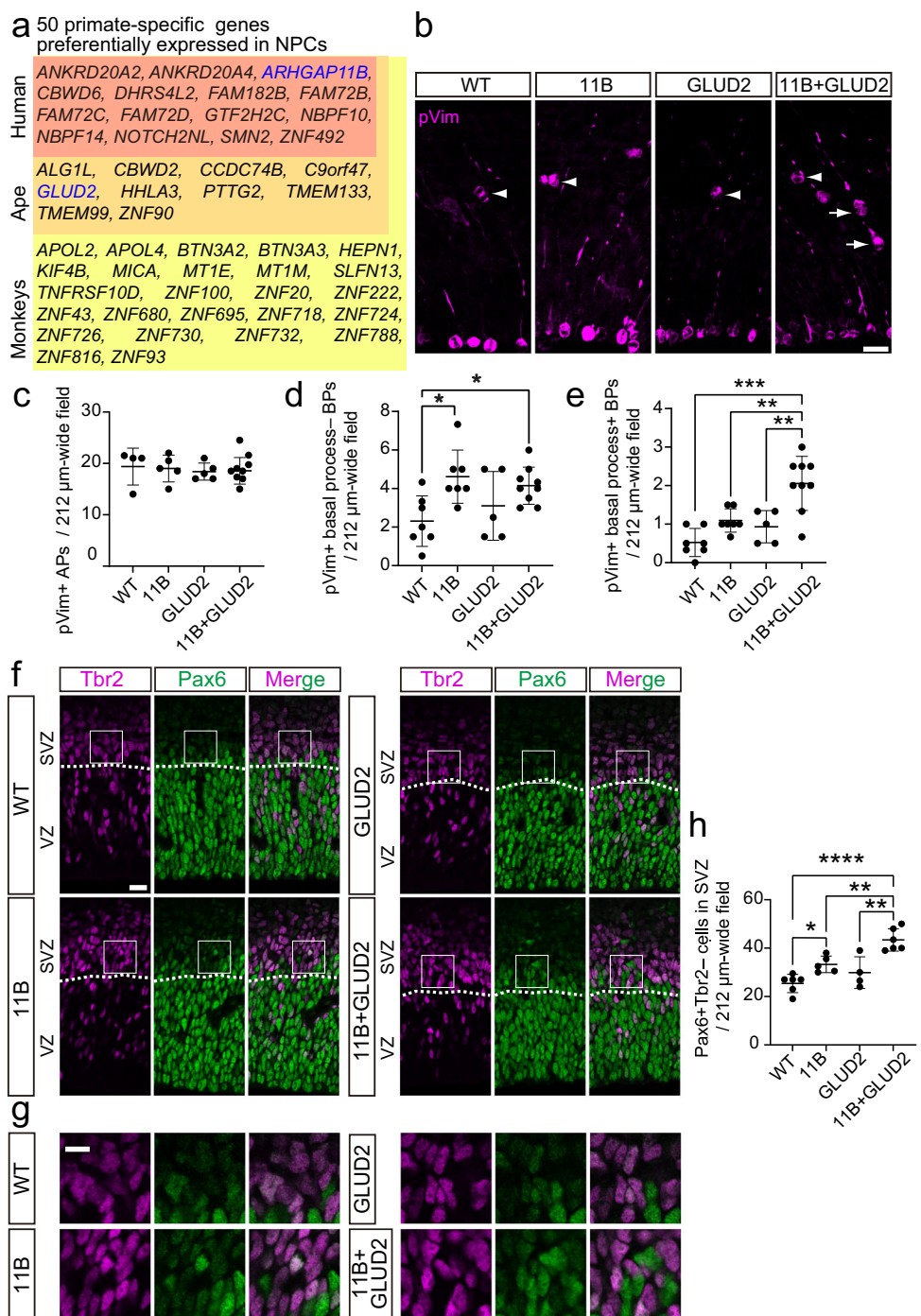

**Fig. 1 | Ape-specific GLUD2 enhances the increase by human-specific ARH-GAP11B in bRG abundance, but not that in bIP abundance. a** Primate-specific genes that are preferentially expressed in NPCs, grouped according to their first appearance in either monkeys, apes, or human (data from Florio et al.[8]). **b** Immunofluorescence of E14.5 wildtype (WT), ARHGAP11B-transgenic (11B), GLUD2-transgenic (GLUD2) and ARHGAP11B + GLUD2 double-transgenic (11B + GLUD2) mouse neocortex for phosphovimentin (pVim, magenta). Arrows indicate mitotic BPs with basal and/or apical process (bRG); arrowheads indicate mitotic BPs without basal and/or apical process (bIPs). **c**–**e** Quantification of pVim+ APs (**c**), pVim+ bIPs (**d**) and pVim+ bRG (**e**), using immunostained cryosections obtained as in **b**. *n* = 4 (WT), 5 (11B), 5 (GLUD2) and 10 (11B + GLUD2) biologically independent embryos in **c**. *n* = 7 (WT), 7 (11B), 5 (GLUD2) and 9 (11B + GLUD2) biologically independent embryos in (**d**, **e**). Each symbol represents one independent embryo. Data are presented as mean values +/- SD; * *p* = 0.159 (WT v.s. 11B in d); * *p* = 0.0495

(WT v.s. 11B + GLUD2 in d); ** *p* = 0.0044 (11B v.s. 11B + GLUD2 in e); **p = 0.0026 (GLUD2 vs. 11B + GLUD2 in e); *** *p* < 0.0001. **f**, **g**, Immunofluorescence of E14.5 wildtype (WT), ARHGAP11B-transgenic (11B), GLUD2-transgenic (GLUD2) and ARHGAP11B + GLUD2 double-transgenic (11B + GLUD2) mouse neocortex for Pax6 (green) and Tbr2 (magenta). Dotted lines indicate the border between ventricular zone (VZ) and subventricular zone (SVZ). Boxed areas in **f** are shown at higher magnification in (**g**). **h** Quantification of Pax6+Tbr2– BPs, using immunostained cryosections obtained as in **f**. *n* = 6 (WT), 5 (11B), 4 (GLUD2) and 6 (11B + GLUD2) biologically independent embryos. Each symbol represents one independent embryo. Data are presented as mean values +/- SD; **p* < 0.049; ***p* < 0.0094 (11B vs. 11B + GLUD2); ** *p* = 0.0013 (GLUD2 vs. 11B + GLUD2); **** *p* < 0.0001. ANOVA followed by Tukey–Kramer tests were used for all analysis. Scale bars: 20 µm in **b**, **f**; 10 µm in **g**. Source data are provided as a Source Data file.

subventricular zone (SVZ), and Tbr2, a marker of neurogenic BPs (notably bIPs)[31,32]. Compared to WT, only ARHGAP11B and ARHGAP11B + GLUD2, but not GLUD2, embryos showed a significant increase in the number of Pax6+Tbr2− cells in the SVZ, with the increase observed in ARHGAP11B + GLUD2 embryos being greater ($p = 0.0094$) than that observed in ARHGAP11B embryos (Fig. 1f–h). Taken together, these results show that GLUD2 enhances the ARHGAP11B-mediated increase in bRG abundance, but not that in bIP abundance.

Consistent with this result, more Pax6+Tbr2− cells were found basally in the cortical wall of E14.5 ARHGAP11B + GLUD2 embryos, while there were no significant differences in the number of Pax6+Tbr2− cells in the VZ and the SVZ (Supplementary Fig. 3a, b). This result prompted us to investigate a possible aRG delamination. However, given that the thickness of the VZ, and the number of abventricular centrosomes, which were visualized by the immunofluorescence for γ-tubulin as previously published[33], were not significantly different among the four genotypes (Supplementary Fig. 3c–e), we conclude that an increased delamination of aRG is not the cause of the increased number of bRG in the ARHGAP11B + GLUD2 embryos.

Since both ARHGAP11B and GLUD2 are mitochondrial protein, we examined a possible effect of ARHGAP11B and GLUD2 expression on mitochondria abundance and morphology. However, when we compared WT and ARHGAP11B + GLUD2 E14.5 neocortex, there were no significant changes in the intensity of the immunofluorescence signal for TOM20, a mitochondrial inner membrane protein, nor in the average length of mitochondria as revealed by TOM20 immunofluorescence, neither in the VZ nor SVZ. These data suggest that the abundance and morphology of mitochondria were not affected by ARHGAP11B and GLUD2 expression (Supplementary Fig. 4a–c).

## The glutamate transporter GLAST, a radial glia marker, is required for the enhancing effect of GLUD2 on the ability of ARHGAP11B to increase bRG abundance

We next sought to explain why GLUD2 is able to enhance the ability of ARHGAP11B to increase bRG abundance but not ARHGAP11B's ability to increase bIP abundance. Since the amino acid substrate of GLUD2 is glutamate, we hypothesized that bRG, but not bIPs, utilize glutamate to promote the ARHGAP11B-induced increase in their proliferation. In this context, it should be noted that in addition to de novo glutamate synthesis, some cell types can take up extracellular glutamate. First, we analyzed two publicly available single-cell/-nuclei RNA sequencing data sets of the fetal human neocortex in the second trimester[34,35] to estimate the glutamate influx into NPCs. In both data sets, the estimated score of glutamate influx (Glutamate_in) was found to be highest for bRG, followed by aRG, and lowest for intermediate progenitors (IPs, which include both bIPs and apical IPs) (Supplementary Fig. 7c, d).

Second, we sought to obtain evidence for glutamate influx into bRG by directly examining whether bRG can take up glutamate. We incubated E14.5 mouse neocortical tissue either with a glutamate containing the stable isotope $^{13}C$ in all 5 carbon atoms ($^{13}C5$-glutamate), or with $^{12}C5$-glutamate at the same concentration as a negative control for 2 h, and then isolated bRG by a previously published procedure[11] (Supplementary Fig. 5) for metabolomic analysis. Upon incubation with $^{13}C5$-glutamate, the isolated bRG were found to contain $^{13}C$-glutamate, and there were no differences in the amount of $^{13}C$-glutamate content between WT and ARHGAP11B + GLUD2 bRG. Thus, bRG in both WT and ARHGAP11B + GLUD2 embryos can take up glutamate (Fig. 2a, Supplementary Fig. 8e, f).

Third, and most relevant, we investigated whether glutamate uptake is crucial for the enhancing effect of GLUD2 on the ability of ARHGAP11B to increase bRG abundance. To this end, we focused on glutamate transporters in the plasma membrane that are enriched in bRG but not bIPs. There are three known glutamate transporters

expressed in the central nervous system (*SLC1A1*, *SLC1A2*/GLT-1, *SLC1A3*/GLAST). Of these transporters, *SLC1A3* shows the highest expression in fetal human bRG (Supplementary Fig. 7a, b). Furthermore, while GLT-1 is mainly expressed in neurons transiently during the neurogenic period in several mammalian species[36–38], GLAST (glutamate aspartate transporter) is a known classical marker protein of radial glia in both embryonic mouse neocortex (Supplementary Fig. 6a, b)[39,40] and fetal human neocortex[32,41]. We therefore explored a possible requirement of GLAST for the increased bRG abundance observed in the neocortex of embryos of the ARHGAP11B + GLUD2 mice, by crossing these mice with previously established GLAST knockout (KO) mice[42]. The elevated number of pVim+ mitotic bRG (Fig. 1e), as well as of Pax6+Tbr2− cells in the SVZ (bRG) (Fig. 1h), in the ARHGAP11B + GLUD2 neocortex at E14.5 was significantly decreased upon GLAST homozygous KO (Fig. 2b–e). These data suggest that glutamate uptake via GLAST is required for the synergistic effect of GLUD2 on the ability of ARHGAP11B to increase bRG abundance.

GLAST homozygous KO did not affect the level of mitotic APs (Fig. 2b, c), nor did it reduce the elevated level relative to WT mice of mitotic bIPs, in the E14.5 ARHGAP11B + GLUD2 mouse embryos (Fig. 2b, c). Accordingly, GLAST homozygous KO did not reduce the abundance of Pax6−Tbr2+ cells (bIPs) in the E14.5 ARHGAP11B + GLUD2 mouse neocortex (Fig. 2d, e). In addition, consistent with previous studies[43], GLAST KO as such, did not affect the abundance of mitotic aRG, bRG, or bIPs in the E14.5 WT mouse neocortex (Supplementary Fig. 6c–f).

We further examined whether glutamate uptake is required to maintain bRG abundance in the fetal human neocortex. Fetal human neocortical tissue at gestation weeks (GW) 12-15 was subjected for 72 h to free-floating tissue (FFT) culture with or without the glutamate uptake inhibitor L-*trans*-pyrrolidine-2,4-dicarboxylic acid (PDC)[44]. There was no statistically significant reduction, upon PDC treatment, in the number of abventricular pVim+TBR2+ cells (mitotic bIPs) when expressed as a ratio to the number of abventricular pVim+TBR2+ cells observed in control experiments when no PDC was added. In contrast, the number of abventricular pVim+TBR2− cells (mitotic bRG) observed upon PDC treatment was significantly decreased when related to that observed upon control treatment (Fig. 2f, g). In addition, PDC treatment significantly decreased the number of SOX2+TBR2− cells (bRG), but not SOX2+TBR2+ cells (bIPs), in the outer subventricular zone (OSVZ) (Fig. 2f, h) when related to the numbers observed upon control treatment. These data show that glutamate uptake is required to maintain bRG abundance.

As to the source of the extracellular glutamate that is taken up by bRG, an obvious primary origin is the blood circulating through the developing brain tissue. However, we wondered to what extent, in addition, the other types of NPCs and the more differentiated cells in the parenchyma of the developing neocortex, i.e. the neurons and macroglial cells, contributed to the extracellular glutamate. Although both neurons and astrocytes are well known to release glutamate, we focused on neurons, given that neurogenesis precedes gliogenesis so that neurons are expected to be more abundant than astrocytes at E14.5. To distinguish between a possible contribution to extracellular glutamate by NPCs vs. non-NPC–type cells, notably neurons, we used two publicly available single-cell/-nuclei RNA sequencing data sets of fetal human neocortex in the second trimester[34,35] to estimate the metabolite flux capabilities of these cells (for details, see Methods). The neurons were found to exhibit higher estimated scores of glutamate efflux (Glutamate_out) than NPCs (Supplementary Fig. 7e). These data suggest that with regard to the contribution to extracellular glutamate by the parenchymal cells of developing neocortex, neurons contribute more than NPCs.

Taken together, our results suggest that in the fetal human neocortex, bRG take up extracellular glutamate, perhaps to some extent

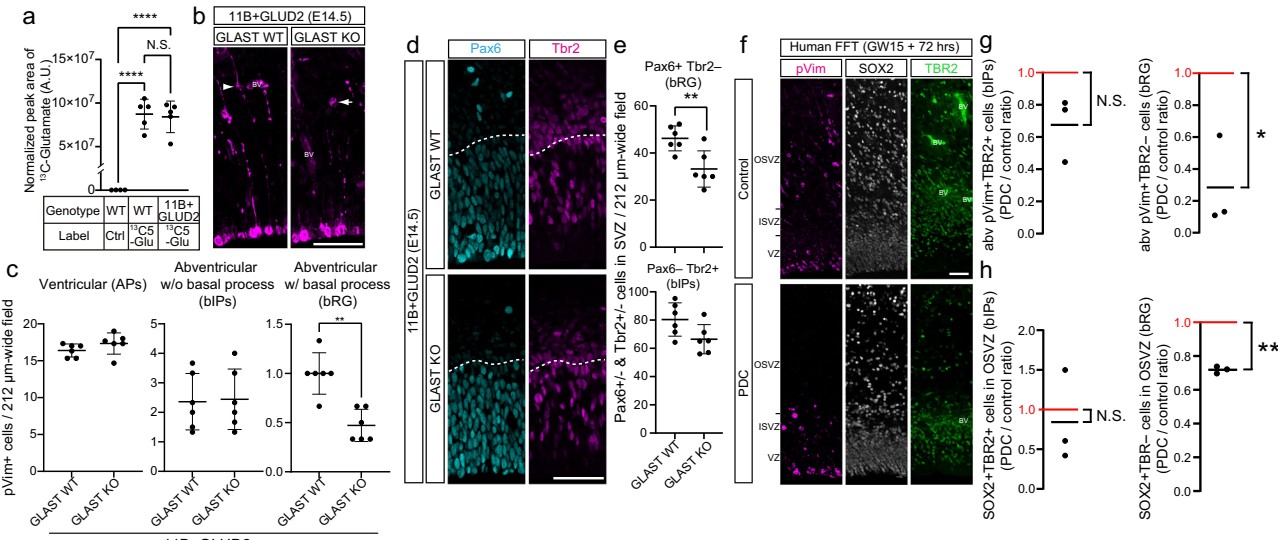

**Fig. 2 | GLAST is required for the enhancement by GLUD2 of the ARHGAP11B-mediated increase in bRG abundance. a** $^{13}C5$-glutamate (Glu) levels in bRG, expressed as normalized peak areas (arbitrary units, A.U.). WT or ARHGAP11B + GLUD2 double-transgenic (11B + GLUD2) mouse neocortical tissue at E14.5 was incubated with 0.5 mM of $^{13}C5$-glutamate (Glu) or unlabeled glutamate (control, Ctrl) for 2 h, followed by isolation of bRG and quantification of $^{13}C5$-glutamate (for details, see Methods). $n = 400,000$ cells from 12 biologically independent embryos per group, examined over 4 independent experiments. Each symbol represents one independent experiment. Data are presented as mean values +/- SD; **** $p < 0.0001$, N.S., not significant. **b** Immunofluorescence of E14.5 ARHGAP11B + GLUD2 double-transgenic and wildtype GLAST (GLAST WT) mouse neocortex and ARHGAP11B + GLUD2 double-transgenic and GLAST knockout (GLAST KO) mouse neocortex for phosphovimentin (pVim, magenta). Arrowhead indicates a mitotic BP with a basal process (bRG); arrow indicates a mitotic BP without a radial process (bIP). BV, blood vessel. **c,** Quantification of pVim+ APs (left), pVim+ bIPs (middle) and pVim+ bRG (right), using immunostained cryosections obtained as in (**b**). $n = 6$ biologically independent embryos per group. Each symbol represents one independent embryo. Data are presented as mean values +/- SD; ** $p = 0.0065$, to-tailed Mann-Whitney test. **d** Immunofluorescence of E14.5 ARHGAP11B + GLUD2 double-transgenic and wildtype GLAST (GLAST WT) mouse neocortex and ARHGAP11B +

GLUD2 double-transgenic and GLAST knockout (GLAST KO) mouse neocortex for Pax6 (cyan) and Tbr2 (magenta). Dotted lines indicate the border between the ventricular zone and subventricular zone. **e** Quantification of Pax6+Tbr2− BPs (bRG, top) and Pax6−Tbr2+ BPs (bIPs, bottom), using immunostained cryosections obtained as in F. $n = 6$ biologically independent embryos per group. Each symbol represents one independent embryo. Data are presented as mean values +/- SD; ** $p < 0.0066$, $t = 3.411$, df=10, two-tailed Student's t test. **f** Fetal human neocortical tissue at GW15 was subjected to FFT culture for 72 h in the absence (Control) or presence of 200 µM of the glutamate uptake inhibitor PDC, followed by immunofluorescence for pVim (magenta), SOX2 (white) and TBR2 (green). **g, h** Quantification of (**g**) abventricular pVim+TBR2+ cells (mitotic bIPs, left) and pVim+TBR2− cells (mitotic bRG, right), and (**h**) SOX2+TBR2+ cells (bIPs, left) and SOX2+TBR2− cells (bRG. right) in the OSVZ, using immunostained cryosections obtained as in **f** and comprising GW12-15. Cell numbers are expressed as a ratio of PDC treatment to control. $n = 3$ biologically independent fetal human neocortices. Each symbol represents one an independent female human neocortex. The red lines at 1.0 indicate the ratio obtained if PDC treatment would have had no effect. Black lines, mean ratio; N.S., not significant; * $p < 0.0483$, $t = 4.385$, df=2, one sample t test; ** $p < 0.0018$, $t = 23.71$, df = 2, one sample t test. Scale bars in **b**, **d**, **f**: 50 µm. Source data are provided as a Source Data file.

released from excitatory neurons, through GLAST to maintain their abundance, which largely reflects the action of ARHGAP11B plus the enhancing effect of GLUD2.

## ARHGAP11B + GLUD2 bRG show increased glutamate-to-aspartate metabolism via the TCA cycle

The previously proposed "three-quarter TCA cycle" concept posits that a partial TCA cycle starting at αKG proceeds only to oxaloacetate (OAA), which is then used for the production of aspartate and further anabolic metabolism[22] (see also Fig. 3a). If so, the increased production of αKG from glutamate due to the action of ARHGAP11B and GLUD2 should result in an increase in the level of aspartate. We therefore investigated the relevant metabolic pathways downstream of glutamate in bRG in the ARHGAP11B and GLUD2 double-transgenic mouse embryos. To trace the metabolic flux of $^{13}C5$-glutamate, WT and ARHGAP11B + GLUD2 double-transgenic neocortical tissue at E14.5 was cultured for 2 h continuously with $^{13}C5$-glutamate, and bRG – and for comparative purposes also aRG – were isolated, using a previously published procedure involving FACS[11], for metabolomic analyses. We first quantified aspartate containing $^{13}C$ in all 4 carbon atoms ($^{13}C4$-aspartate), which can be generated from $^{13}C5$-glutamate through one three-quarter TCA cycle (3/4 TCA cycle) via OAA (Fig. 3a), and aspartate retaining $^{13}C$ in only 2 carbon atoms ($^{13}C2$-aspartate), which can be generated from $^{13}C5$-glutamate via OAA through one full TCA cycle

followed by one 3/4 TCA cycle (Fig. 3a). The ratio of $^{13}C4$-aspartate to $^{13}C5$-glutamate (Fig. 3b, Supplementary Fig. 8a) and of $^{13}C2$-aspartate to $^{13}C5$-glutamate (Fig. 3c, Supplementary Fig. 8a) was significantly increased in ARHGAP11B + GLUD2 bRG compared to WT bRG. In contrast, when we quantified proline containing $^{13}C$ in all 5 carbon atoms ($^{13}C5$-proline), which can also be generated from $^{13}C5$-glutamate (Fig. 3a), the ratio of $^{13}C5$-proline to $^{13}C5$-glutamate was decreased in the ARHGAP11B + GLUD2 bRG compared to WT bRG (Fig. 3d, Supplementary Fig. 8a). When we quantified glutamine containing $^{13}C$ in all 5 carbon atoms ($^{13}C5$-glutamine), which can also be generated from $^{13}C5$-glutamate (Fig. 3a), the ratio of $^{13}C5$-glutamine to $^{13}C5$-glutamate was not significantly different between ARHGAP11B + GLUD2 bRG and WT bRG (Fig. 3e, Supplementary Fig. 8a). In addition, the fractional enrichment of $^{13}C4$- and $^{13}C2$-aspartate in the total isotopologues of aspartate (Supplementary Fig. 8c), and the actual amount of $^{13}C4$- and $^{13}C2$-aspartate detected (Supplementary Fig. 8e) showed significant increases in ARHGAP11B + GLUD2 bRG compared to WT bRG.

We further explored whether two alternative pathways known to generate aspartate from glutamate were affected by ARHGAP11B and GLUD2 expression. One such pathway is a TCA-independent pathway to produce aspartate through a reductive carboxylation by which glutamate is converted into aspartate[45]. The other pathway is the so-called γ-aminobutyric acid (GABA) shunt, in which glutamate is first converted to GABA, which is then converted to aspartate via partial

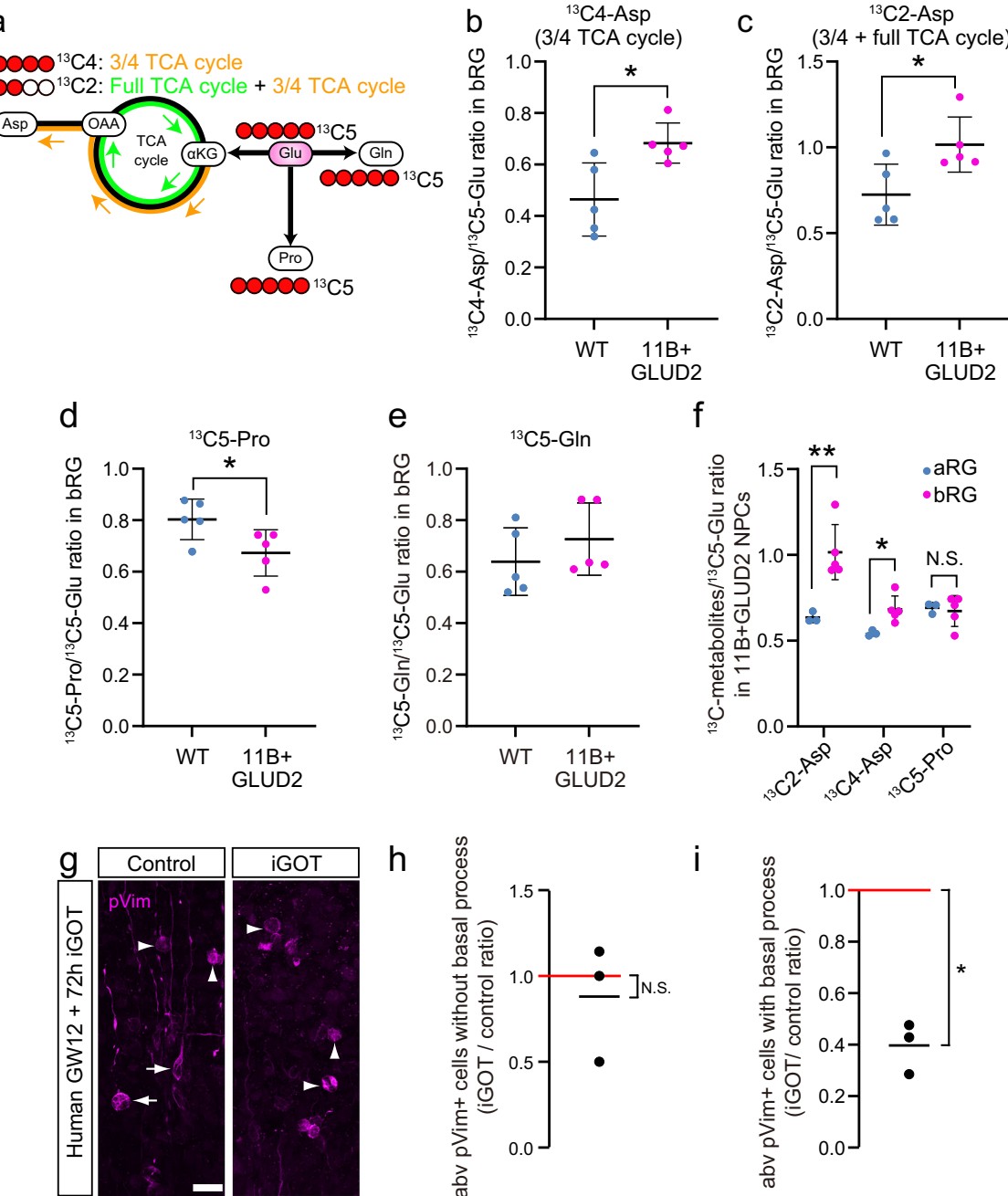

**Fig. 3 | Increased aspartate production from glutamate via the TCA cycle in ARHGAP11B + GLUD2 bRG, and its relevance for bRG proliferation in fetal human neocortex. a** Schematic illustration of selected aspects of glutamate metabolism. [13]C5-glutamate (Glu) can be metabolized to [13]C5-glutamine (Gln), to [13]C5-proline (Pro), and to either [13]C4-aspartate (Asp) generated via αKG and oxaloacetate (OAA) through one three-quarter (¾) TCA cycle (orange) or [13]C2-aspartate generated through one full TCA cycle (green) followed by one three-quarter (¾) TCA cycle. [13]C atoms are indicated by red circles. **b–f** WT or ARHGAP11B + GLUD2 double-transgenic (11B + GLUD2) mouse neocortical tissue at E14.5 was incubated with 0.5 mM of [13]C5-glutamate for 2 h, followed by isolation of bRG and metabolome analysis (for details, see Methods). n = 500,000 cells from 15 biologically independent embryos per group, examined over 5 independent experiments. Each symbol represents one independent experiment. Data are presented as mean values +/- SD. * $p = 0.0165$, $t = 3.021$, df=8 in **b**; * $p = 0.026$, $t = 2.727$, df=8 in **c**; * $p = 0.041$, $t = 2.433$, df=8 in d; * $p = 0.025$, $t = 2.972$, df=6; ** $p = 0.0077$, $t = 3.936$, df=6; all tests are two-tailed Student's t test. **b–e**, Values of [13]C4-aspartate (Asp) (**b**), [13]C2-aspartate (Asp) (**c**), [13]C5-proline (Pro) (**d**) and [13]C5-glutamine (Gln)

(**e**), each expressed as a ratio to that of [13]C5-glutamate (Glu), in bRG from WT (blue) and ARHGAP11B + GLUD2 double-transgenic (magenta) mouse neocortex at E14.5. **f**, Values of [13]C2-aspartate (Asp), [13]C4-aspartate (Asp) and [13]C5-proline (Pro), each expressed as a ratio to that of [13]C5-glutamate (Glu), in aRG (blue) and bRG (magenta) from ARHGAP11B + GLUD2 double-transgenic mouse neocortex at E14.5. **g** Fetal human neocortical tissue at GW13 was subjected to FFT culture for 72 h in the absence (Control) or presence of 10 μM of the GOT1/2 inhibitor iGOT, followed by immunofluorescence for pVim (magenta). **h, i**, Quantification of (**h**) abventricular pVim+ cells without radial processes (mitotic bIPs) and (**i**) pVim+ cells with radial processes (mitiotic bRG), using immunostained cryosections obtained as in **g** and comprising GW12-15. Cell numbers are expressed as a ratio of iGOT treatment to control. $n = 3$ biologically independent fetal human neocortices. Each symbol represents one independent female human neocortex. The red lines at 1.0 indicate the ratio obtained if iGOT treatment would have had no effect. Black lines, mean ratio. * $p = 0.0089$, $t = 10.54$, df=2, one sample t test. Error bars, SD; N.S., not significant; Scale bars in **g**: 10 μm. Source data are provided as a Source Data file.

usage of the TCA cycle[46]. However, neither $^{13}C3$-aspartate (which would be the metabolite of $^{13}C5$-glutamate in the case of its reductive carboxylation), $^{13}C4$-GABA (which would be the metabolite of $^{13}C5$-glutamate if the GABA shunt were used) nor $^{13}C2$-GABA showed a significant increase in ARHGAP11B + GLUD2 bRG compared to WT bRG; rather, there actually was a reduction in the amount of $^{13}C4$-GABA detected in the ARHGAP11B + GLUD2 bRG (Supplementary Fig. 8a, c, e).

Taken together, when various metabolic pathways downstream of glutamate are compared between ARHGAP11B + GLUD2 bRG and WT bRG, ARHGAP11B + GLUD2 bRG seem to exhibit a greater degree of glutamate-to-αKG conversion followed by either one 3/4 TCA cycle or one full TCA cycle followed by one 3/4 TCA cycle, to produce more aspartate at the expense of proline production from glutamate. As aspartate is critical for cell proliferation[47], this provides an explanation for the increase in bRG abundance by ARHGAP11B and the enhancement of this increase by GLUD2 (Fig. 1).

When the ratios of $^{13}C4$-aspartate, $^{13}C2$-aspartate and $^{13}C5$-proline to $^{13}C5$-glutamate were compared between aRG and bRG isolated from ARHGAP11B + GLUD2 double-transgenic E14.5 mouse neocortical tissue, for both $^{13}C4$- and $^{13}C2$-aspartate, but not for $^{13}C5$-proline, this ratio was found to be higher in bRG than aRG (Fig. 3f; for primary data, see Supplementary Fig. 8a–f). This corroborates the importance of aspartate production from glutamate for the increase in bRG abundance in ARHGAP11B + GLUD2 E14.5 mice, and suggests that bRG are more receptive than aRG to the changes in the metabolic pathways downstream of glutamate induced in the neocortical tissue of ARHGAP11B + GLUD2 E14.5 mice.

### Aspartate production from oxaloacetate is required for bRG proliferation in fetal human neocortical tissue ex vivo

We examined whether aspartate production from OAA is required for the proliferative capacity of bRG in the fetal human neocortex. To this end, fetal human neocortical tissue ex vivo at gestational weeks (GW) 12-15 was subjected for 72 h to FFT culture with or without iGOT[48], an inhibitor of glutamic-oxaloacetic transaminases (GOT1/2), which are critical for OAA-to-aspartate conversion. The tissue was then subjected to pVim immunofluorescence to quantitate mitotically active BPs. There was no statistically significant change in the number of abventricular pVim+ cells without apical and/or basal processes (i.e. mitotic bIPs) upon iGOT treatment when expressed as a ratio to the number of such cells in control tissue (Fig. 3g, h). In contrast, the number of abventricular pVim+ cells with apical and/or basal processes (i.e. mitotic bRG) observed upon iGOT treatment was decreased by more than half when expressed as a ratio to the number of such cells observed upon control treatment (Fig. 3g, i). These data indicate that OAA-to-aspartate conversion is required for bRG proliferation in fetal human neocortical tissue ex vivo.

### αKG alone is sufficient to increase BP abundance and can rescue the reduction in bRG abundance upon inhibition of ARHGAP11B function

We next examined whether fueling the TCA cycle by the end product of glutaminolysis, αKG, is alone sufficient to increase bRG abundance. To this end, WT mouse hemispheres at E13.5 were cultured for 48 h in the presence or absence of a plasma membrane-permeable αKG ester, ETaKG, which is converted intracellularly to αKG[49]. While there were no changes in the number of ventricular pVim+ cells (i.e. mitotic APs, Fig. 4a, b), the number of abventricular pVim+ cells with a basal process (i.e. mitotic bRG, Fig. 4a, d) or without a basal process (i.e. mitotic bIPs, Fig. 4a, c) was significantly increased upon ETaKG treatment compared to control. Thus, αKG supplementation is sufficient to increase the abundance of BPs, including that of bRG.

Finally, we examined whether αKG supplementation can rescue the phenotype caused by a disruption of ARHGAP11B function in the fetal human neocortex, which implies disruption of any enhancement of this

function by GLUD2. Because both *ARHGAP11B* and *GLUD2* arose by gene duplication, genome editing or knockdown specific to *ARHGAP11B* or *GLUD2* are extremely challenging, and there are no small molecules known that selectively inhibit the functions of the corresponding proteins. Therefore, we took advantage of a previously developed dominant-negative variant of ARHGAP11B, ARHGAP11A220[21], to disrupt the function of ARHGAP11B and hence any ability of GLUD2 to enhance this function. Fetal human neocortical tissue at GW13 was subjected to ex vivo electroporation with either a plasmid containing the *ARHGAP11A220* cDNA or a control plasmid, together with a GFP-expressing plasmid to detect transfected cells and their progeny, and then cultured for 72 h, in the case of *ARHGAP11A220* transfection in the presence or absence of ETaKG. No changes were observed, in the three experimental conditions analyzed, in the percentage of GFP+ cells that were PCNA+ in the VZ (mostly APs, Fig. 4e, f) and PCNA+TBR2+ in the SVZ (bIPs, Fig. 4e, h). In contrast, the reduction in the percentage of the GFP+ cells in the SVZ that were PCNA+TBR2− (bRG) observed upon expression of ARHGAP11A220 compared to control was rescued by the ETaKG treatment (Fig. 4e, g). This indicates that the elevated level of bRG in fetal human neocortex, that is, decreased when the function of ARHGAP11B is disrupted, can be rescued by αKG supplementation. On the likely assumption that GLUD2, which is endogenously present in fetal human neocortical NPCs (Supplementary Fig. 1), contributes to the ability of ARHGAP11B to maintain the elevated level of bRG in fetal human neocortex, the complete rescue of AHRGAP11B function with regard to bRG abundance further implies that αKG supplementation also rescues the contribution of GLUD2 to the above ability of ARHGAP11B.

## Discussion

The present study demonstrates the functional synergy of a human-specific gene and an ape-specific gene (Supplementary Fig. 9a) in the NPC amplification that is crucial for the evolutionary expansion of the human neocortex. Specifically, while the ape-specific GLUD2 alone does not increase bRG abundance, it enhances the bRG amplification by ARHGAP11B.

bRG take up glutamate from their microenvironment via the glutamate transporter GLAST (Supplementary Fig. 9b). This likely promotes bRG proliferation because it provides an additional source for the generation of αKG via GLUD2, which is added to the αKG generated by the glutaminolysis promoted by ARHGAP11B[21]. The αKG is metabolized in a "three-quarter TCA cycle" to oxaloacetate, which is then converted to aspartate, as previously hypothesized[22]. Aspartate is an essential amino acid for de novo nucleotide synthesis, and is known to be a limiting factor for cell proliferation[47]. Therefore, the functional synergy of ARHGAP11B and GLUD2 promotes a pro-proliferative metabolism in bRG by increasing the production of aspartate.

Functional synergy of genes can create nonlinear phenotypic changes during evolution[50,51]. Interestingly, a comparison of hominid brain size as predicted from endocranial volume revealed a steep, greater-than-linear increase that started around two million years ago[52]. The present study provides evidence for a key role of the functional synergy between genes in the evolutionary expansion of the human neocortex. In the ancestors of humans, the duplication and modification of *ARHGAP11B* generated a novel protein that exploited the existence of the gene *GLUD2*, that emerged earlier in the ancestors of apes and humans (Supplementary Fig. 9a) to increase, via concerted changes in metabolism (Supplementary Fig. 9b), the abundance of neuronal progenitors needed for the expansion of the neocortex on the human evolutionary lineage.

## Methods

### Ethics

All animal experiments were performed in accordance with the German Animal Welfare legislation (Germany) and Act on the Protection of

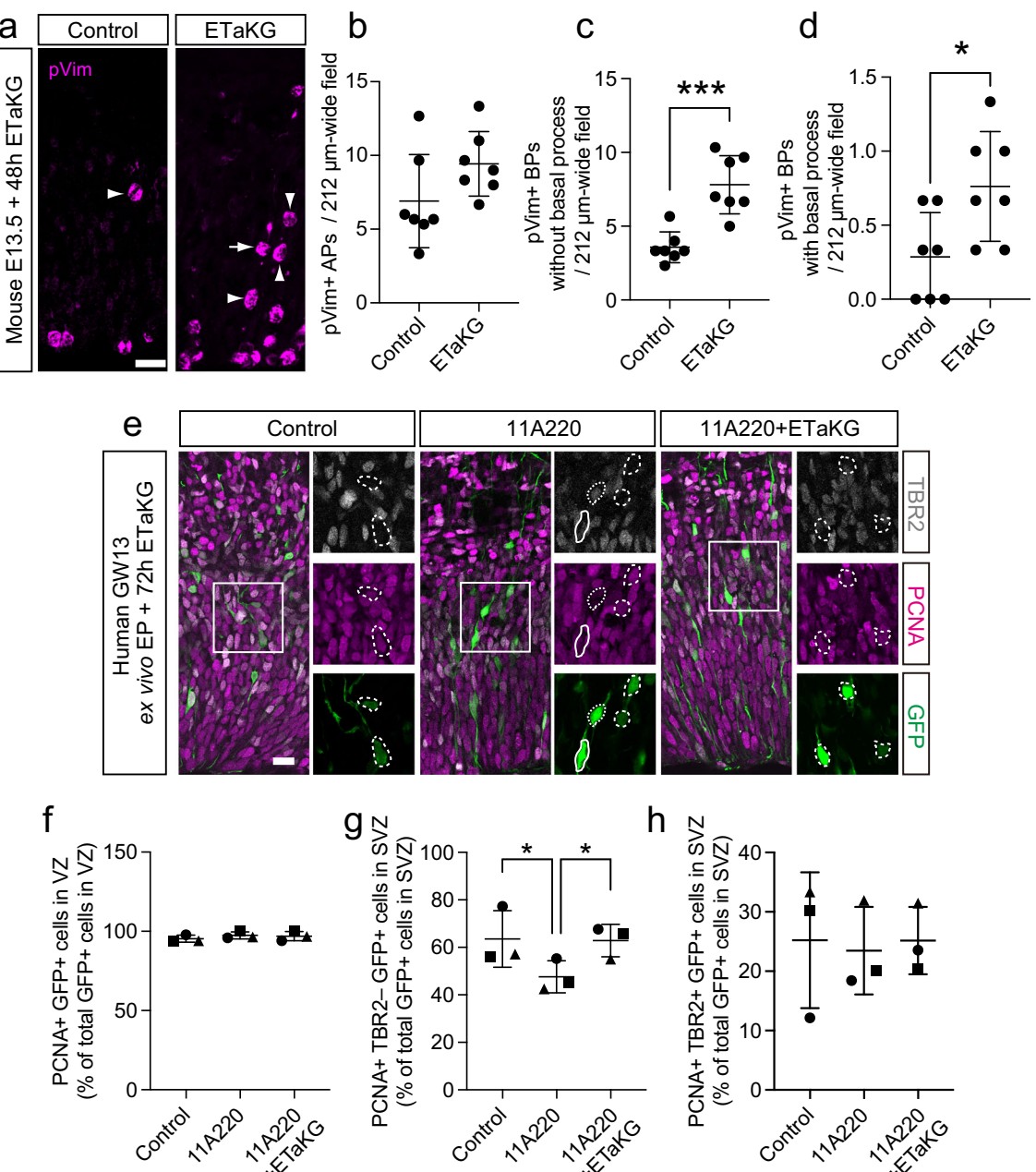

**Fig. 4 | Extracellularly supplied αKG alone suffices to increase bRG abundance in embryonic wildtype mouse neocortex and rescues the reduction in bRG abundance upon ARHGAP11B inhibition in fetal human neocortex.**
**a** Immunofluorescence of E13.5 mouse neocortical tissue, incubated for 48 h without (Control) or with 1 mM ETaKG, for phosphovimentin (pVim, magenta). Arrow indicates a mitotic BP with a basal (bRG); arrowheads indicate mitotic BPs without radial processes (bIPs). **b–d** Quantification of pVim+ APs (**b**), pVim+ bIPs (BPs without radial processes, **c**) and pVim+ bRG (BPs with radial processes, **d**), using immunostained cryosections obtained as in (**a**). n = 7 biologically independent embryos per group. Each symbol represents one independent embryo. Data are presented as mean values +/- SD; * p = 0.0215, t = 2.641, df=12; *** p = 0.0003, t = 5.023, df=12; all tests are two-tailed Student's t test. **e** GW13 human neocortical tissue electroporated with a plasmid encoding GFP, together with either an empty vector (Control) or a human ARHGAP11A220-expressing plasmid (11A220), was incubated for 48 h in the absence (Control, 11A220) or presence (11A220) of 0.5 mM ETaKG (11A220+ETaKG). Upon the three types of treatment, the tissue was

subjected to immunofluorescence for PCNA (magenta), TBR2 (white), and GFP (green). Boxed areas in the triple-immunofluorescence images are shown at higher magnification at the right of each image, and show the individual immuno-fluorescence signals. Selected PCNA + TBR2–GFP+ cycling BPs (bRG) are outlined by dashed white lines; one selected PCNA–TBR2+ GFP+ newborn neuron is outlined by dotted white lines; selected PCNA+TBR2+GFP+ cycling BPs (bIPs) are outlined by solid white lines. **f–h** Quantification of the percentage of GFP+ cells in the VZ that are PCNA+ (**f**), and of GFP+ cells in the SVZ that are PCNA+TBR2– (bRG, **g**) or PCNA+TBR2+ (bIPs, **h**), upon control, 11A220 and 11A220+ETaKG treatment, using immunostained cryosections obtained as in **e**. n = 3 biologically independent fetal human neocortices. Each symbol represents one independent female human neocortex. Data are presented as mean values +/- SD; * p = 0.038 (Control v.s. 11A220); * P = 0.043 (11A220 v.s. 11A220+ETaKG), ANOVA followed by Tukey–Kramer tests. Scale bars in **a** and **e**: 20 μm. Source data are provided as a Source Data file.

Animals Used for Scientific or Educational Purposes (Finland). All procedures regarding the animal experiments were approved by the Governmental agencies (Landesdirektion Sachsen, Germany; Aluehallintovirasto, Finland) and overseen by the Institutional Animal Welfare Officer(s). The license numbers concerning the experiments with mice are: Untersuchungen zur Neurogenese in Mäuseembryonen TVV2015/05 and ESAVI/15112/2021 (in utero electroporation) and 24−9168.24-9/2012-1 (tissue collection without prior in vivo experimentation).

## Mice
ARHGAP11B-transgenic mice[29] and GLUD2 BAC transgenic mice (GLUD2-521177)[30] at E14.5 were used to generate heterozygous ARHGAP11B and GLUD2 double transgenic mice. To generate GLAST (Slc1a3) knockout (KO) in the ARHGAP11B and GLUD2 double transgenic background, we used homozygotic GLAST:CreERT2 mice as a GLAST KO mouse, in which CreERT2 cassette is knocked in the GLAST loci[42]. We crossed homozygous ARHGAP11B + GLUD2 double-transgenic mice with homozygous GLAST KO mice to obtain heterozygous ARHGAP11B + GLUD2 + GLAST KO triple-transgenic mice. These triple-transgenic mice were crossed with each other to obtain homozygous ARHGAP11B + GLUD2 + GLAST KO mice, which carry two copies of ARHGAP11B, two copies of GLUD2 and zero copy of GLAST. The homozygous ARHGAP11B + GLUD2 + GLAST KO mice were then crossed with homozygous GLAST KO mice to obtain mice heterozygous for ARHGAP11B + GLUD2 and homozygous for GLAST KO, which carry one copy of ARHGAP11B, one copy of GLUD2 and zero copy of GLAST. C57BL/6 N mice were used as WT for the experiments used ARHGAP11B + GLUD2 double transgenic mice, otherwise C57BL/6 J mice were used. Mice were maintained in pathogen-free conditions at the animal facility of the Max Planck Institute of Molecular Cell Biology and Genetics (MPI-CBG) or University of Helsinki (UH), Finland. All mice were housed in the animal facility of MPI-CBG and UH under state-of-the-art husbandry conditions and with a normal light/dark cycle. All experiments were performed in accordance with German and Finnish animal welfare legislation and were overseen by the Institutional Animal Welfare Officer. The mice were kept under 12 h light/dark cycle.

The mouse brain tissue we used was embryonic and of an early stage of neurogenesis, and so we did not distinguish between sex, because at this stage of embryonic neurogenesis any influence or association of sex on the results was highly unlikely.

## Human tissues
Fetal human brain tissues (GW12-15) were obtained from the Helsinki University Hospital, with approval by the ethics committee of the Hospital district of Helsinki and Uusimaa (HUS/1170/2021). For immunohistochemistry, fetal human brain tissues were dissected in PBS and subjected to the tissue culture experiments or were immediately fixed with 4% PFA overnight at 4 °C, and then washed with PBS followed by 30% sucrose in PBS for immunohistochemistry. The tissues were embedded in OCT compound (Sakura Finetek USA) and frozen using liquid nitrogen. The frozen samples were kept at − 80 °C for long-term storage.

## Plasmids
pCAGGS-ARHGAP11B, -ARHGAP11A220 and -EGFP have been described previously[6,21]. Full length human GLUD1 was first cloned into pCR-Blunt II-TOPO, and then subcloned into pCAGGS to obtain pCAG-GLUD1. pCAG-GLUD2 was made by Synbio Technologies, NJ, USA.

## Antibodies
Antibodies used in this study were as follows; anti-ARHGAP11B (mouse IgG1, 3758-A37-5, MPI-CBG, 1:100)[21], anti-γ-tubulin (mouse IgG, T6557, Sigma, 1:500), anti-GFP (goat IgG, MPI-CBG, 1:500), anti-GFP (chicken IgY, GFP-1020, Aves, 1:500), anti-GLAST (guinea pig IgG,

GLAST-GP-Af1000, Frontier institute, Japan, 1:200), anti-GLUD1/2 (rabbit IgG, ab166618, abcam, 1:5000), anti-Pax6 (rabbit IgG, 901301, BioLegend, 1:200), anti-PCNA (mouse IgG, CBL407, Millopore), anti-PH3 (rat, ab10543, Abcam), anti-pVim (mouse IgG, D076-3, MBL., 1:200), anti-SOX2 (goat IgG, AF2018, R&D Systems, 1:500), anti-Tbr2 (rabbit IgG, ab23345, abcam, 1:400), anti-TOM20 (mouse IgG, ab56783, Abcam, 1:200), anti-TOM20 (rabbit IgG, ab78547, Abcam, 1:1000), anti-chicken IgY-Alexa Fluor 488 (donkey, 703-545-155, Jackson ImmunoResearch), anti-goat IgG-Alexa Fluor 488 (donkey, A11055, ThermoFisher Scientific), anti-goat IgG-Alexa Fluor 647 (donkey, A21447, ThermoFisher Scientific), anti-goat IgG-Alexa Fluor 647 (donkey, 705-605-147, Jackson ImmunoResearch), anti-guinea pig IgG-Aexa Fluor 647 (donkey, 706−605-148, Jackson ImmunoResearch), anti-mouse IgG-Alexa Fluor 488 (donkey, A21202, ThermoFisher Scientific), anti-mouse IgG-Alexa Fluor 555 (donkey, A31570, ThermoFisher Scientific), anti-mouse IgG-Alexa Fluor 647 (donkey, A31571, ThermoFisher Scientific), anti-mouse IgG-Cy3 (donkey, 715-165-151, Jackson ImmunoResearch), anti-mouse IgG-HRP (donkey, 715-035-151, Jackson ImmunoResearch), anti-rabbit IgG-Alexa Fluor 488 (donkey, A21206, ThermoFisher Scientific), anti-rabbit IgG-Alexa Fluor 488 (donkey, 711-545-152, Jackson ImmunoResearch), anti-rabbit IgG-Alexa Fluor 647 (donkey, A31573, ThermoFisher Scientific), anti-rabbit IgG-HRP (donkey, 711-035-152, Jackson ImmunoResearch). The dilution of all Alexa-labeled antibodies was 1:500; all Cy3-labeled antibodies were 1:200; all HRP-labeled antibodies were 1:5000.

Anti-GLUD2 antibody was produced using human GLUD2-specific peptide (PTAEFQDSISGA)[53] in rabbit by Innovagen AB, Sweden. The specificity of rabbit polyclonal anti-GLUD2 (#13924.13), which was affinity purified by Protein-G and the GLUD2-specific peptide, was tested by Innovagen with ELISA using the GLUD2-specific peptide and GLUD1-specific peptide (PTAEFQDRISGA) (Supplementary Fig. 1g). The specificity was also tested by immunoblot of cell lysate in which GLUD1 or GLUD2 were overexpressed (see below) (Supplementary Fig. 1e, f).

## Cell transfection and immunoblot
COS-7 cells were transfected with indicated plasmids (1–2.5 μg per transfection) by Amaxa nucleofector (Lonza) according to the manufacturer's protocols. Cells were cultured in cell growth medium (DMEM (Lonza) supplemented with 10% FBS and 1x Penicillin-Streptomycin (Invitrogen)) at 37 °C under an atmosphere of humidified 5% $CO_2$ and 95% air. Total cell lysates were made one day after the transfection, and processed for immunoblot. SDS-PAGE was performed using Novex Bis-Tris Gels (4%–12%; ThermoFisher Scientific) in NuPAGE MOPS SDS RunningBuffer (ThermoFisher Scientific) according to the manufacturer's protocol. After the electrophoresis, proteins were transferred onto PVDF membrane (Immobilon-P, Merck) in NuPAGE Transfer Buffer (ThermoFisher Scientific) for 2 h. The membranes were then incubated in TBST (Tris-buffered saline containing 0.1% Tween 20) containing 5% BSA for 1 h at r.t. with gentle shaking, followed by incubation with the indicated primary antibodies in TBST overnight at 4 °C with gentle shaking. The membranes were washed with TBST and then incubated with appropriate HRP-conjugated secondary antibodies in TBST for 1 h at r.t. with gentle shaking. Finally, membranes were developed with Pierce ECL Western Blotting Substrate (ThermoFisher Scientific). Exposure time was varied for best visibility. Images were acquired using PG-BOX Chemi XX6 (SYNGENE).

## In utero electroporation (IUE) of embryonic mouse neocortex
IUE was performed as previously described[54]. Briefly, pregnant mice with E13.5 embryos were anesthetized with isofluorane, then subcutaneously injected with the analgesic (0.1 ml, Metamizol, 200 mg/kg). The embryos were injected intraventricularly with a solution containing 0.1% Fast Green (Sigma) in sterile 154 mM NaCl, 2 μg/μl of the pCAGGS plasmid (empty vector, ARHGAP11B and GLUD2) and 0.3 μg/μl of the EGFP vector using a glass micropipette,

followed by electroporation (24 V, five 50 msec pulses with 950 msec intervals). Mice received Metamizol for one day via drinking water (1.33 mg/ml) after the surgery. Mice were sacrificed by cervical dislocation and embryos were harvested 20–24 h post-electroporation, and the embryonic brains were dissected and fixed with 4% PFA for immunohistochemistry (see below).

## Mouse cerebral hemisphere culture

Cerebral hemispheres were dissected from E13.5 mouse embryos and placed in the hemisphere rotation (HERO) culture according to the protocol[55], with minor modifications. Hemispheres were cultured with slice culture medium (SCM; Neurobasal medium (Invitrogen) supplemented with 20 mM L-glutamine (Invitrogen), 1x Penicillin-Streptomycin (Invitrogen) or 100 µg/ml Streptomycin (Merck), 1x N2 supplement (Invitrogen), 1x B27 supplement (Invitrogen), 0.1 mM HEPES-NaOH (pH 7.3) and 10% Knockout Serum Replacement (Thermo Fisher Scientific)). The SCM contained 1 mM 5-Ethyl 1-(3-(trifluoromethyl)benzyl) 2-oxopentanedioate (ETaKG; SML1743, Sigma) or DMSO (Sigma), followed by incubation for 48 h at 37 °C under an atmosphere of humidified 5% $CO_2$, 40% $O_2$ and 55% $N_2$. The hemispheres were then fixed with 4% PFA for immunohistochemistry (see below).

## Human tissue electroporation and free-floating tissue (FFT) culture

Ex vivo electroporation of the fetal human neocortical tissue was performed as described previously[21]. Neocortical tissue was placed onto a spoon-shaped anode filled with sterile PBS. The mixture of the following plasmids in a solution containing 0.1% Fast Green (Sigma), 5% glycerol and PBS, was added onto the apical surface of the tissue. The plasmid mixture consists of pCAGGS-EGFP at 1 µg/µl together with pCAGGS empty vector at 2 µg/µl for control or pCAGGS-ARHGAP11A220 at 2 µg/µl. The cathode was placed on top of the tissue, and electroporations were performed (40 V, five 50 msec pulses, 950 msec intervals), thus, the plasmid DNA entered the cells in the apical-to-basal direction. After the electroporation, the tissue was washed in PBS and placed into a rotating flask with 1.5 or 2 mL SCM containing the following reagents: i) 0.5 mM ETaKG or DMSO as a control, ii) 200 µM PDC or PBS as a control, iii) 10 mM iGOT (N-(4-chlorophenyl)−4-(1H-indol-4-yl)piperazine-1-carboxamide; BD01294031, BLDPharm) or DMSO as a control. The tissues were incubated as FFT culture at 37 °C for 3 days in a humidified atmosphere of 5% $CO_2$, 60% $O_2$, and 35% $N_2$ as previously described[55,56]. The hemispheres were then fixed with 4% PFA for immunohistochemistry (see below).

## Immunofluorescence

Immunostaining was performed as previously described[57]. Embryonic mouse brains and fetal human brain tissues were cut into 14 µm-thick or 40 µm-thick cryosections, respectively. Antigen retrieval was performed by incubating the sections in 0.01 M sodium citrate buffer for 60 min at 70 °C, followed by incubation for 20 min r.t. Sections were further permeabilized with 1% (wt/vol) Triton X-100 in PBS for 30 min and quenched remaining PFA with 2 mM glycine in PBS for 30 min, followed by blocking with a solution containing 0.2% (wt/vol) gelatin, 300 mM NaCl and 0.3% (wt/vol) Triton X-100 in PBS (TX buffer). Sections were incubated overnight at 4 °C with primary antibodies that were diluted in the TX buffer. After washing with PBS, the sections were incubated with the appropriate secondary antibodies in the TX buffer for 60 min at r.t., followed by washing with PBS.

## Image acquisition and quantification

The fluorescent images were acquired using LSM700, LSM 880, and LSM980, with 40x and 63x objectives. The images were taken as tile scans and stitched using the ZEN 3.3 software (Zeiss). All cell countings and other quantifications were performed in standardized microscopic fields using ZEN software and Fiji 1.54, as indicated in the figure legends. Any pVim+ cell 30 µm from the apical surface and any GFP+ cell lacking apical contact and 30 µm from the apical surface was counted as a BP. The fluorescent signal intensity of ARHGAP11B and GLUD2 was measured by Fiji. The signal intensity was normalized to set the maximum intensity of each fluorescent signals to 1 (Fig.S1I).

Determination of zones in the developing neocortex was done according to the previous publications[57–59]. The VZ and SVZ in embryonic mouse neocortex was determined based on the Tbr2+ and Pax6+ cell density and morphology of nuclei. The VZ consists of dense Pax6+ cells with relatively elongated nuclear morphology and less Tbr2+ cells. The SVZ consists of a band of dense Tbr2+ cells with round nuclear morphology. The VZ, ISVZ, and OSVZ in the fetal human neocortex were determined based on the Tbr2+ and Sox2+ or PCNA+ cell density. The VZ consists of dense Sox2+ or PCNA+ cells and less Tbr2+ cells. The ISVZ consists of the dense Tbr2+ cells and less Sox2+ or PCNA+ cells. The OSVZ consists of scattered Tbr2+ cells and Sox2+ or PCNA+ cells. Since the morphology of nuclei could be affected by FFT culture condition, we did not consider the morphology of nuclei as one of the criteria for fetal human tissues.

## $^{13}$C5-Glutamate labeling of mouse neocortical tissues

Lateral neocortical tissues from E14.5 WT and ARHGAP11B + GLUD2 double-transgenic embryos were incubated in SCM with 500 µM unlabeled glutamate (49621, Sigma) or 500 µM $^{13}$C5-glutamate (604860, Sigma) for 2 h at 37 °C in a humidified atmosphere of 5% $CO_2$, 60% $O_2$ and 35% $N_2$. The concentration of glutamate and the length of the incubation time were optimized based on pilot experiments to obtain the sufficient number of cells after the FACS, and achieve sufficient conversion of $^{13}$C5-glutamate to downstream metabolites, before reaching a steady state, for the analysis of rate and metabolic pathways in the cells investigated[60].

## Isolation of aRG and bRG from mouse neocortex by FACS

Isolation of aRG and bRG was performed as previously described[11]. Single-cell suspensions were prepared incubated embryonic mouse neocortical tissue using the MACS Neural Tissue Dissociation kit containing papain (Miltenyi Biotec) following the manufacturer's instruction. Cell-surface staining of prominin-1 (Prom-1) and GLAST was performed on the cell suspensions with rat 13A4 APC-conjugated antibody (1:50, eBioscience, Clone 13A4, #17-1331-81, RRID:AB_823120) and with anti-EAAT1/GLAST-1/SLC1A3 PE-conjugated antibody (1:10, Novus Biologicals, #NB100-1869PE). Next, fluorescence-activated cell sorting (FACS) was performed using a 5-laser-BD FACSAria Fusion sorter (Becton Dickinson Biosciences; see Supplementary Fig. 5). Cell suspensions were first washed twice with Hanks' Balanced Salt Solution (HBSS) before being subjected to FACS. Gates were applied as follows. First, a P1 gate was set on the SSC-A/FSC-A dot-plot, to identify live cells based on size and shape. Next, the P1 fraction was restricted by setting a P2 gate on the FSC-W/FSC-H dot-plot to select single cells. Out of the P2 population, single dot-plots were created for SSC-A/PE (linear/log2, yellow-green laser, 561 nm) and SSC-A/APC (linear/log2, red laser, 640 nm) to visualize the fluorescence intensities of Prom-1-APC and GLAST-PE, respectively. Voltage parameters were set based on an unstained control, and subsequently maintained for FACS. Next, the GLAST+/Prom-1+ gate was restrictively set and maintained through FACS to acquire aRG and the GLAST+/Prom-1− gate was restrictively set and maintained through FACS to acquire bRG, both of which were sorted at 4 °C into 100 µl HBSS in Eppendorf tubes. Sorted aRG and bRG were briefly centrifuged at 1000x g for 5 min, cell pellets were rapidly frozen in liquid nitrogen and then kept at −80 °C until further analysis.

## Targeted metabolic flux analysis

After FACS sorting of cells to ice-cold PBS, the metabolites were extracted from pelleted cells with 400 µl of cold extraction solvent (Acetonitrile:Methanol:Water; 40:40:20, Thermo Fischer Scientific).

Subsequently, samples were three times vortexed 2 min and sonicated for 1 min followed by centrifugation at 14000 rpm at 4 °C for 5 min. Next the samples were centrifuged, the usupernatant transferred to an evaporation tube and evaporated to dry under nitrogen stream. Samples were reconstituted in 40ul extraction buffer (40:40:20; Acetonitrile:Methanol:Water) and transferred to LC-MS vials. Supernatants were transferred into polypropylene tubes and placed into a Nitrogen gas evaporator and dried samples were suspended with 40 μl of extraction solvent (Acetonitrile:Methanol:Water; 40:40:20) and vortexed for 2 min and transfer into HPLC glass autosampler vials. 2 μl of sample injected with Thermo Vanquish UHPLC coupled with Q-Exactive Orbitrap quadruple mass spectrometer equipped with a heated electrospray ionization (H-ESI) source probe (Thermo Fischer Scientific). A SeQuant ZIC-pHILIC (2.1×100 mm, 5-μm particle) column (Merck) was used for chromatographic separation. The gradient elution was carried out with a flow rate of 0.100 ml/min with using 20 mM ammonium hydrogen carbonate, adjusted to pH 9.4 with ammonium solution (25%) as mobile phase A and acetonitrile as mobile phase B. The gradient elution was initiated from 20% Mobile phase A and 80% of mobile phase B and maintain till 2 min., followed by 20% Mobile phase A gradually increasing up to 80% till 17 min., then 80% to 20% Mobile phase A decrease in 17.1 min. and maintained up to 24 min. The column oven and auto-sampler temperatures were set to 40 ± 3 °C and 5 ± 3 °C, respectively. Following setting were used for MS: polarity switching; resolution of 35,000, the spray voltages: 4250 V for positive and 3250 V for negative mode; the sheath gas: 25 arbitrary units (AU); the auxiliary gas: 15 AU; sweep gas flow 0; capillary temperature: 275 °C; S-lens RF level: 50.0. Instrument control was operated with the Xcalibur software (Thermo Fischer Scientific). The peak integration was done with the TraceFinder 5.1 software (Thermo Fischer Scientific) using confirmed retention times standardized with library kit MSMLS-1EA (Merck). 13 C isotopologolues were analyzed with change of m/z (m + 1, m + 2 etc.). The data quality was monitored throughout the run using a pooled QC sample prepared by pooling 5 μL from each suspended samples and interspersed throughout the run as every 10th sample. The metabolite data was checked for peak quality, % relative standard deviation (RSD) and carryover. Each metabolite peak area was normalized to the total cell number analyzed, and calculated as a ratio to the normalized peak areas of $^{13}C5$-glutamate or as a fraction of the sum of all isotopologues (e.g. all isotopologues of glutamate are $^{13}C0$-, $^{13}C1$-, $^{13}C2$-, $^{13}C3$-, $^{13}C4$- and $^{13}C5$-glutamate) of each metabolite (i.e. fractional enrichment).

### Gene expression analysis by RT−qPCR

Total RNA was isolated from E14.5 wild type, ARHGAP11B transgenic, GLUD2 transgenic and ARHGAP11B + GLUD2 double transgenic mouse neocortex using the RNAeasy Micro Kit (Qiagen) according to the manufacturer's instructions. cDNA was synthesized using the Maxima first-strand cDNA synthesis kit for RT−qPCR (Thermo Scientific). qPCR was performed using the FastStart essential DNA green master (Roche) and LightCycler" 96 Instrument (Roche). Specific primers for *Arhgap11a* (forward primer 5'-TCCGTCAGTCCGTCAGAAGA-3', reverse primer 5'-CTGCGTCACCAAAGATTGCC-3'), *ARHGAP11B* (forward primer 5'- GGAGTAGCACAGAGA-3', reverse primer 5'- TGAGAATAAAATGGACAGCAGCA), *Glud1* (forward primer 5'- AGGGGACTCTTGGGAACTGGT-3', reverse primer 5'- GACTCCAAACAGGGAGCCCAA-3'), *GLUD2* (forward primer 5'-TGGCTTTTCCCAGCACAATCAG-3', reverse primer 5'- AGCGTGTCCCAGACTCATCC-3') and *Gapdh* (forward primer 5'-TGAAGCAGGCATCTGAGGG-3', reverse primer 5'-CGAAGGTGGAAGAGTGGGAG-3') were used for RT-qPCR analysis.

### Single-cell/nuclei RNA sequencing data analysis

Two published single cell/nuclei RNA sequencing data sets of fetal human neocortex[34,35] were used. The Raw count data and associated metadata from Polioudakis and colleagues were downloaded from the CoDEx viewer (http://solo.bmap.ucla.edu/shiny/webapp/), while the data from Uzquiano and colleagues were retrieved from the Single Cell Portal (https://singlecell.broadinstitute.org/single_cell/study/SCP1756). The normalized count data and provided cell type labels were used for analysis.

The graph neural network model-based single-cell flux estimation analysis (scFEA) was applied to estimate cell-wise metabolic flux using glutaminolysis modules. The default parameters were utilized, except for the original glutamate_in module, which was divided into glutamate_in (*SLC1A1*, *SLC1A2*, *SLC1A3*, *SLC1A5*, *SLC1A6*, *SLC1A7*) and glutamate_out (*SLC17A6*, *SLC17A8*, *SLC17A7*) modules based on the transporter functions. The code of scFEA and module genes were available on GitHub (https://github.com/changwn/scFEA).

### Statistical analysis

Data were analyzed with Excel (Microsoft, Redmond, WA), Statcel3 (OMS, Japan), MYSTAT (Systat Software, CA) and GraphPad Prism 9 (GraphPad Software). Mouse embryos and human fetal tissues were randomly selected for all experiments. No data were excluded from the quantifications. If the tissue samples appeared to be severely damaged or dead, mouse or human tissue samples were excluded from further analyses. No predetermination of sample sizes was carried out because our research is an exploratory study. Sample collection and quantification were done blindly for the following image analysis: Fig. 1, Fig. 2c, e, g, h, Fig. 3h, i, Fig. 4, Supplemental Fig. 2, Supplemental Fig. 3, Supplemental Fig. 4, Supplemental fig. 6. Supplemental Fig. 1c and were done blindly. Metabolomics was done blindly by the core facility. Re-analyses of single cell and bulk transcriptome data were not done blindly since we need to know cell types to run the downstream analyses. Supplemental Fig. 1l was not done blindly since we needed to know that the cells are in the OSVZ. The normality of the data were tested by the Shapiro-Wilk test. The following tests were used: One-way ANOVA followed by the Tukey-Kramer multiple comparison test for Fig. 1c−e, h, Fig. 2a, Fig. 4f−h, Supplementary Fig. 2b, Supplementary Fig. 3a-c, e; Unpaired Student's *t*-test for Fig. 2c(left and middle panels), e, Fig. 3b−d, Fig. 4b−d, Supplementary Fig. 4b, c, Supplementary Fig. 8; Mann-Whitney test for Fig. 2c (right panel), Fig. 3e, f; One sample *t*-test for Fig. 2g, h, Fig. 3h, i; One-way ANOVA followed by Dunnett's multiple comparison test for Supplementary Fig. 1a, b; One-way ANOVA on ranks (the Kruskal−Wallis test) followed by the Dunn's multiple comparison test for Supplementary Fig. 6b−d. df means degree of freedom. Please also see the Source Data.

### Reporting summary

Further information on research design is available in the Nature Portfolio Reporting Summary linked to this article.

## Data availability

All data are available in the manuscript or the supplementary material. Human samples used in this study are available only for researchers/institutes that are listed on the patient informed form. Source data are provided with this paper.

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

## Acknowledgements

We apologize to all researchers whose work could not be cited because of space limitations. We are indebted to Magdalena Götz for generously providing the GLAST KO mice. We thank Beatrice Öhman and her team at the Helsinki University Hospital for human sample collection; the members of the Namba lab and the Huttner lab for useful discussions; M. Bespalov for the pilot experiments for iGOT; E. Huttu, S. Lågas, I. Gómez Lozano and C. Haffner for technical assistance; Y. Hiraoka for technical advices; the services and facilities of the University of Helsinki as well as the Biocenter Finland for the outstanding support provided, especially the Biomedicum Flow Cytometry, Biomedicum Imaging Unit, FIMM Metabolomics/Lipidomics/Fluxomics Unit and HiLIFE Laboratory Animal Centre Core Facility for their technical support, and the services and facilities of the Max Planck Institute of Molecular Cell Biology and Genetics for their technical support, notably J. Peychl and his team of the Light Microscopy Facility, J. Helppi and his team of the Biomedical Services (BMS), and J. Jarrells and J Amairani Hernández Méndez and their team of the Cell Technology facility; and the Transgenic Core Facility. Supported by the Research Council of Finland (340179, 351966) (T.N.), ERA-NET NEURON (MEPIcephaly) (T.N. and P.C.), the Sigrid Jusélius Foundation (T.N.), HiLIFE Fellow program at the University of Helsinki (T.N.), Cancer Foundation Finland (T.N.) and the Brain Science Foundation (T.N.), the NOMIS Foundation (S.P.), a grant from ERA-NET NEURON (MicroKin) (W.B.H.), the Max Planck Society (S.P. and W.B.H.), and partially supported by the profiling area Understanding the Human Brain (UHBRAIN) of PROFI 6 Competitive funding to strengthen university research profiles, granted by the Research Council of Finland to the University of Helsinki (336234) (T.N.).

## Author contributions

Conceptualization: L.X, W.B.H., T.N.; formal analysis: L.X.; A.I.N, H-C.Z., T.N.; investigation: L.X., V.G., A.I.N., H-C.Z., R.N., K.R., T.N.; resources: M.A., K.T., P.C., O.H., S.P.; writing–original draft: L.X., T.N.; writing–review and editing: L.X., W.B.H., T.N. with comments from A.I.N, H-C.Z., S.P.; project administration and supervision: W.B.H., T.N.

## Funding

## Competing interests

O.H. occasionally serves on advisory boards for Bayer AG and Gedeon Richter and has designed and lectured at educational events for these companies. The remaining authors declare no competing interests.
