## [Peer Review File · Nature Communications]

Functional synergy of a human-specific and an ape-specific metabolic regulator in human neocortex developmentREVIEWER COMMENTS

Reviewer #1 (Remarks to the Author):

Summary: In this very interesting manuscript, the authors characterize the impact of two genes, GLUD2 and ARGHAP11B on bRG abundance and glutamate metabolism. The authors find that, in mice, GLUD2 enhances the increase in bRGs seen with the introduction of ARGHAP11B. They find that the increase in bRGs caused by the presence of GLUD2 and ARGHAP11B is dependent on GLAST, a glutamate transporter. They also find that bRGs isolated from ARGHAP11B+/GLUD2+ mouse cortex show greater flux from glutamate to aspartate through the TCA cycle. Furthermore, they find that the exogenous addition of aKG can rescue the decrease in bRGs caused by ARGHAP11B inhibition, suggesting that ARGHAP11B's impact on bRG abundance is mediated by its role in increasing TCA cycle intermediates.

Comments/Revisions:

Figure 1/S1:

1. It may be useful to include the expression of GLUD2/ARGHAP11B in the various cell types of interest (bRG, aRG, bIPs) as opposed to just neurons vs bRGs (currently this is in Fig S1A).

Figure 2:

1. What are the units for Figure 2A?

Figure 3/S5:

1. If possible, I think including abundance of the metabolites in each condition as well as the amount that are labeled could help inform the tracing data.

2. I think some explanation of why 2 hours was chosen for the trace may be helpful – has the trace reached a steady state at this point in time?

3. I don't fully understand the normalization to proline for the aRGs. The aspartate data looks similar between bRGs and aRGs in the supplement (although I'm not certain as the scales are different) and the proline data looks different between bRGs and aRGs, suggesting that the proline may potentially be driving the effect seen here. As the focus seems to be on an increase in aspartate generation from glutamate in bRGs as compared to a decrease in proline generation, I think perhaps including the reasoning here for this normalization strategy, changing the wording to focus on proline, or changing to a glutamate normalization would make sense.

4. It is not clear to me whether TCA cycle flux (and its offshoots, including aspartate and its byproducts) is higher in the ARGHAP11B/GLUD2+ bRGs, or whether it is specific to the $\frac{3}{4}$ TCA cycle/aspartate catabolism pathway. I think softening the language around this hypothesis by mentioning that it could be higher TCA cycle flux generally as opposed to specifically a $\frac{3}{4}$ TCA cycle or showing flux data for some other TCA cycle intermediates to demonstrate that there is a specifically higher routing through aspartate as opposed to general TCA flux could be helpful or contribute to the story.

5. If GLUD2+ mouse bRGs and ARGHAP11B+ mouse bRGs were traced (as opposed to just GLUD2+/ARGHAP11B+ mouse bRGs) with labeled glutamate and metabolites in the TCA cycle/downstream of it were analyzed, I think it may strengthen the claims of the authors about this hypothesis: "bRG take up glutamate from their microenvironment via the glutamate transporter GLAST (Fig. S6b). This likely promotes bRG proliferation because it provides an additional source for the

generation of α KG via GLUD2, which is added to the α KG generated by the glutaminolysis promoted by ARHGAP11B.” If I understand this hypothesis correctly, it might also help disentangle the role of GLUD2 vs. ARGHAP11B in increasing bRG abundance. However, I don’t think this is a necessary revision, but may bolster this hypothesis stated in the conclusion.

Methods:

1. I’m confused by this sentence: “We crossed homozygous ARHGAP11B+GLUD2 double transgenic mice, which carry homozygous GLAST KO, with homozygote GLAST KO mice to obtain heterozygous ARHGAP11B+GLUD2 with homozygous GLAST KO.”

Reviewer #2 (Remarks to the Author):

The manuscript explores the roles of ARHGAP11B and GLUD2 individually and together in neocortex development in a human model system. The importance of ARHGAP11B and GLUD2 is showcased using advanced microscopy techniques. Metabolic alterations are studied using stable isotope-labeled compounds, namely, $^{13}\text{C}_5$ -glutamate. The fate of the ^{13}C -carbons through various pathways is shown using select metabolites.

As expected, the ratios of $^{13}\text{C}_x/^{13}\text{C}_y$ are essentially one to one for most of the metabolites shown, suggesting that there's no disturbance in metabolism. However, the authors point out that there might be slightly more $^{13}\text{C}_4$ Asp than $^{13}\text{C}_5$ -Asp, suggesting that the genetic variants accelerate glutaminolysis. Unfortunately, the metabolites shown for this pathway are limited, and intermediate products are not explored sufficiently to support the claim that aspartate is derived from glutamine via the TCA cycle. Most troublesome is the fact that flux data are not expressed or interrogated in regard to normal ^{12}C metabolites. Also alternative, TCA independent pathway are not investigated or not shown. Therefore, actual changes in modification of metabolic pathway cannot be determined with sufficient certainty.

Additionally, the statement that aspartate is an essential amino acid is incorrect, as essentially all human cells can synthesize aspartate, making it a non-essential amino acid. The study shows that Aps is derived from Glu, suggesting that the cells studied can make it.

Given the stunning morphological changes, it's unlikely that the minor change in aspartate is a driving factor. It is more plausible that the genetic alteration causes changes in neocortex development that subsequently affect the glutamine to aspartate ratio. In general, small changes in genes and gene expressions that are important for development can have drastic (>2-fold) effects on metabolic flux. Another weakness in the flux analysis is the fact that the aspartate pathway is not quantified, and from the presented data, it's impossible to determine what fraction of glutamate is actually going towards glutamine, proline, or aspartate. Thus, while this reviewer agrees that there might be a slide shift in $^{13}\text{C}_x$ Asp, the data present are insufficient to support the strong statements made.

In the opinion of this reviewer, the statement that the genetic alteration induces glutamate to aspartate flux via the TCA cycle is not sufficiently supported by the data as presented especially since

transaminases and other common housekeeping enzymes can produce the observed flux.

Reviewer #3 (Remarks to the Author):

In this original work, to help identify the mechanism of action of ARHGAP11B, authors focused on primate-specific metabolic regulators in mitochondria in neural progenitor cells. GLUD2 was identified as acting synergistically to ARHGAP11B. Extracellular glutamate is also required to increase bRG abundance (including uptake via GLAST).

The authors performed transgenic expression in mouse neocortex (mouse lines as well as IUE). The combination of GLUD2 and ARHGAP11B leads to an increased production of bRGs. The Glast KO reduced bRG production, and in human fetal brain tissue a glutamate uptake inhibitor also specifically reduced bRG. RNAseq data (previously published) suggests bRGs have high glutamate influx and neurons high efflux. Metabolic pathways were investigated downstream of glutamate in bRG in the ARHGAP11B and GLUD2 double-transgenic mouse embryos. Significantly increased aspartate levels (and decreased proline levels) were identified in ARHGAP11B+GLUD2 bRG compared to WT bRG (and higher in bRG than aRG) suggesting a greater degree of glutamate-to- α KG conversion. α KG supplementation was then found to be sufficient to increase the abundance of BPs, including that of bRG in mouse cortex. α KG supplementation also rescued the phenotype caused by a disruption of ARHGAP11B function in fetal human neocortex.

The results are interesting describing a functional synergy between two proteins in the evolutionary expansion of the human neocortex and changes in metabolism, which have arisen during evolution.

Comments and questions

Does ARHGAP11B interact physically with GLUD2 (and /or other enzymes in the glutaminolysis or TCA pathway)? Does this happen only in the mitochondria or to some extent also in the cytosol? Are there other proteins (not necessarily primate-specific) which may positively influence the function of ARHGAP11B in these pathways?

GLAST is mentioned but might other SLCs also be expected to be important to enhance production or maintenance of bRGs?

Figure 1 f,g,h and page 7: What about overall numbers of Pax6 + cells? It is important to show this data and a binning analysis. Do double transgenics have a thinner VZ?

ARHGAP11B / GLUD2 immunohistochemistry – are these two proteins only expressed in a proportion of bRGs (c.f. Sox2 labelling Supp Fig 1)? Signals in more apical regions seem quite large and diffuse, as well as more intense, why is this? Also, how is the distribution in soma versus basal processes – it would be good to see co-staining with a mitochondrial marker.

Is there an impact of transgenic expression on cells in mouse brain (transgenes expressed in APs as well as BPs), do cells change in size?, is there an impact on number and morphology of mitochondria?

Page 8, conclusion: Taken together, our results suggest that in fetal human neocortex, bRG take up extracellular glutamate, ... through GLAST to maintain their abundance, which largely reflects the action of ARHGAP11B plus the enhancing effect of GLUD2'. There is only indirect evidence to suggest that glutamate uptake is required to maintain bRG abundance - it is not confirmed that there is an effect on proliferation, and what about the possibility that there is an impact on bRG production from aRG? It is interesting that aRGs in the double transgenic model also show metabolic changes compared to WT in the double transgenic (Supp Fig 5b), even if it is less /different than the effect on bRGs. This impact on abundance (compared to production) needs to be shown more conclusively.

Even if aspartate is stated to be essential for proliferation, increased proliferation of bRGs needs to be shown here to support the conclusions.

In addition, as shown in Fig 4a-d, bIPs as well as bRGs are increased with added α KG supplementation and there is also a tendency for increase in aRGs. Does the double transgenic therefore increase BP production?

Minor

In several experiments, bRGs or aRGs are purified for the analyses. It would be important to show data convincing the reader of the quality of this purification.

Authors say on page 6, transgenes are similar to the expression pattern observed in fetal human neocortex (Fig. S1c-i) – but they don't show this.

GLUD2 sounds like glutamate receptor delta, it would be important to incorporate the full name in the abstract.

Fig 1: it is written that 'the number of pVim+ adventricular cells without basal and/or apical processes, considered to be mitotic bIPs,.... the number of pVim+ adventricular cells with basal and/or apical processes, considered to be mitotic bRG', these two categories overlap? Need to be clearer.

Fig 2: bRG can uptake glutamate, but we are only told in Supp Fig 4 (RNA seq analysis), how this compares to bIPs, I would suggest to move this information up in the manuscript, otherwise testing glutamate uptake in bIPs and aRGs using the same methods as for bRGs, could become a necessity. aRGs are not mentioned in the text, although this information is incorporated in Supp Fig 4. It would be good to mention aRGs as well.

How does Glast expression compare in the cell types? The SLC which is GLAST should be indicated clearly in the legend.

Fig 4e-h, why were PCNA+ cells analysed specifically?

Was the sex of the mouse embryos assessed or of the human fetal cortex known?

The English is good, but another read through by a native English speaker could be good

Response to Reviewers

Reviewer #1

Reviewer's comments:

Summary: In this very interesting manuscript, the authors characterize the impact of two genes, GLUD2 and ARGHAP11B on bRG abundance and glutamate metabolism. The authors find that, in mice, GLUD2 enhances the increase in bRGs seen with the introduction of ARGHAP11B. They find that the increase in bRGs caused by the presence of GLUD2 and ARGHAP11B is dependent on GLAST, a glutamate transporter. They also find that bRGs isolated from ARGHAP11B+/GLUD2+ mouse cortex show greater flux from glutamate to aspartate through the TCA cycle. Furthermore, they find that the exogenous addition of aKG can rescue the decrease in bRGs caused by ARGHAP11B inhibition, suggesting that ARGHAP11B's impact on bRG abundance is mediated by its role in increasing TCA cycle intermediates.

Authors' Response:

We greatly appreciate that the Reviewer finds our manuscript "very interesting", and thank the Reviewer for this concise summary of our study.

Reviewer's comments:

Comments/Revisions:

Figure 1/S1:

1. It may be useful to include the expression of GLUD2/ARGHAP11B in the various cell types of interest (bRG, aRG, bIPs) as opposed to just neurons vs bRGs (currently this is in Fig S1A).

Authors' Response:

As suggested by the Reviewer, we now added the expression of both *GLUD2* and *ARGHAP11B* in aRG, in addition to bRG and neurons, to the revised Fig. S1a,b as suggested by the Reviewer. However, the previous transcriptomic study from the Huttner lab did not analyze the bIP population, and we therefore cannot add TPM values for bIPs to the revised Fig. S1a,b.

Reviewer's comments:

Figure 2:

1. What are the units for Figure 2A?

Authors' Response:

The units of the normalized peak area of ¹³C-glutamate are arbitrary units (A.U.). The ordinate of Fig. 2a is now labeled accordingly, and this is explained in the corresponding figure legend.

Reviewer's comments:

Figure 3/S5:

1. If possible, I think including abundance of the metabolites in each condition as well as the amount that are labeled could help inform the tracing data.

Authors' Response:

We agree with the Reviewer's comment. As requested by the Reviewer, we have now added the abundance of all metabolites as values of normalized peak area, again expressed as arbitrary units, in the new Fig. S8e and f. In addition, we have carried out additional analyses to show the fractional enrichment of isotopologues (for example, for glutamate the isotopologues are ¹³C0-, ¹³C1-, ¹³C2-, ¹³C3-, ¹³C4- and ¹³C5-glutamate) in bRG (new Fig. S8c) and in aRG (new Fig. S8d).

Reviewer's comments:

2. I think some explanation of why 2 hours was chosen for the trace may be helpful – has the trace reached a steady state at this point in time?

Authors' Response:

We have chosen a 2-hour incubation period based on pilot experiments in order to obtain a sufficient number of isolated cells after the FACS step for parallel replicates in the tracing analyses. We found that for all cell types analyzed, 2 hours are sufficient for the labeled carbons from glutamate to be detected in aspartate and the intermediate metabolites. We did not analyze the metabolites at a timepoint when the cells have reached a steady state, as this will not reflect the metabolism rate and paths taken in bRG in different conditions if all labeled glutamate has already been converted into the final products. This reasoning is now explained in the revised Methods section.

Reviewer's comments:

3. I don't fully understand the normalization to proline for the aRGs. The aspartate data looks similar between bRGs and aRGs in the supplement (although I'm not certain as the scales are different) and the proline data looks different between bRGs and aRGs, suggesting that the proline may potentially be driving the effect seen here. As the focus seems to be on an increase in aspartate generation from glutamate in bRGs as compared to a decrease in proline generation, I think perhaps including the reasoning here for this normalization strategy, changing the wording to focus on proline, or changing to a glutamate normalization would make sense.

Authors' Response:

We appreciate the Reviewer's comment. We have revised Fig. 3f to compare the ratio of $^{13}\text{C}2\text{-Asp}$, $^{13}\text{C}4\text{-Asp}$ and $^{13}\text{C}5\text{-Pro}$ to $^{13}\text{C}5\text{-Glu}$ between aRG and bRG in the *ARHGAP11B+GLUD2* transgenic embryos. As shown in the revised Fig. 3f, the ratio of $^{13}\text{C}2\text{-Asp}$ and $^{13}\text{C}4\text{-Asp}$ to $^{13}\text{C}5\text{-Glu}$ is higher in bRG than aRG. However, there are no significant difference in the ratio of $^{13}\text{C}5\text{-Pro}$ to $^{13}\text{C}5\text{-Glu}$ between aRG and bRG (revised Fig. 3f).

Reviewer's comments:

4. It is not clear to me whether TCA cycle flux (and its offshoots, including aspartate and its byproducts) is higher in the *ARGHAP11B/GLUD2+* bRGs, or whether it is specific to the $\frac{3}{4}$ TCA cycle/aspartate catabolism pathway. I think softening the language around this hypothesis by mentioning that it could be higher TCA cycle flux generally as opposed to specifically a $\frac{3}{4}$ TCA cycle or showing flux data for some other TCA cycle intermediates to demonstrate that there is a specifically higher routing through aspartate as opposed to general TCA flux could be helpful or contribute to the story.

Authors' Response:

The Reviewer has raised a fair point. Since we have found that both $^{13}\text{C}2\text{-Asp}$ and $^{13}\text{C}4\text{-Asp}$ were increased in the *ARHGAP11B+GLUD2* transgenic bRG, this indicates – as anticipated by the Reviewer – that flux via both a $\frac{3}{4}$ TCA cycle and a full TCA cycle followed by a $\frac{3}{4}$ TCA cycle was increased. As requested by the Reviewer, we have softened the language here to avoid overstating the case.

Reviewer's comments:

5. If *GLUD2+* mouse bRGs and *ARGHAP11B+* mouse bRGs were traced (as opposed to just *GLUD2+/ARGHAP11B+* mouse bRGs) with labeled glutamate and metabolites in the TCA cycle/downstream of it were analyzed, I think it may strengthen the claims of the authors about this hypothesis: “bRG take up glutamate from their microenvironment via the glutamate transporter GLAST (Fig. S6b). This likely promotes bRG proliferation because it provides an additional source for the generation of αKG via *GLUD2*, which is added to the αKG generated by the glutaminolysis promoted by *ARHGAP11B*.” If I understand this hypothesis correctly, it might also help disentangle the role of *GLUD2* vs. *ARGHAP11B* in increasing bRG abundance.

However, I don't think this is a necessary revision, but may bolster this hypothesis stated in the conclusion.

Authors' Response:

We thank the Reviewer for this comment and the suggestion to strengthen our hypothesis. We agree with the Reviewer that tracing labeled glutamate in *GLUD2*+ bRG and *ARHGAP11B*+ bRG would bolster our hypothesis. At the same time, we greatly appreciate that the Reviewer does not think that such experiments are a necessary revision, as tracing labeled glutamate in two additional single transgenic mouse lines would take a very long time and would greatly delay publication of our manuscript.

Reviewer's comments:

Methods:

1. I'm confused by this sentence: "We crossed homozygous ARHGAP11B+GLUD2 double transgenic mice, which carry homozygous GLAST KO, with homozygote GLAST KO mice to obtain heterozygous ARHGAP11B+GLUD2 with homozygous GLAST KO."

Authors' Response:

We agree with the Reviewer that the previous sentence was confusing, and have now explained the crossing of mice in greater detail in the Methods section, as follows: "*We crossed homozygous ARHGAP11B+GLUD2 double transgenic mice with homozygous GLAST KO to obtain heterozygous ARHGAP11B+GLUD2+GLAST KO triple transgenic mice. These triple transgenic mice were crossed with each other to obtain homozygous ARHGAP11B+GLUD2+GLAST KO mice, which carry two copies of ARHGAP11B, two copies of GLUD2 and zero copy of GLAST. The homozygous ARHGAP11B+GLUD2+GLAST KO mice were then crossed with homozygous GLAST KO mice to obtain mice heterozygous for ARHGAP11B+GLUD2 and homozygous for GLAST KO, which carry one copy of ARHGAP11B, one copy of GLUD2 and zero copy of GLAST.*"

Reviewer #2

Reviewer's comments:

The manuscript explores the roles of ARHGAP11B and GLUD2 individually and together in neocortex development in a human model system. The importance of ARHGAP11B and GLUD2 is showcased using advanced microscopy techniques. Metabolic alterations are studied using stable isotope-labeled compounds, namely, $^{13}\text{C}_5$ -glutamate. The fate of the ^{13}C -carbons through various pathways is shown using select metabolites.

As expected, the ratios of $^{13}\text{C}_x/^{13}\text{C}_y$ are essentially one to one for most of the metabolites shown, suggesting that there's no disturbance in metabolism. However, the authors point out that there might be slightly more $^{13}\text{C}_4$ Asp than $^{13}\text{C}_5$ -Asp, suggesting that the genetic variants accelerate glutaminolysis. Unfortunately, the metabolites shown for this pathway are limited, and intermediate products are not explored sufficiently to support the claim that aspartate is derived from glutamine via the TCA cycle. Most troublesome is the fact that flux data are not expressed or interrogated in regard to normal ^{12}C metabolites. Also alternative, TCA independent pathway are not investigated or not shown. Therefore, actual changes in modification of metabolic pathway cannot be determined with sufficient certainty.

Authors' Response:

We would like to address the Reviewer's comments as follows. First, of the several pathways that generate Asp from Glu, only the pathway through glutaminolysis and the TCA cycle (i.e. Glu to αKG to Asp through the TCA cycle) transfers four carbon atoms of Glu to Asp via a $\frac{3}{4}$ TCA cycle, and 2 carbon atoms of Glu to Asp by a full TCA cycle followed by a $\frac{3}{4}$ TCA cycle. Therefore, the detection of $^{13}\text{C}_4$ -Asp or $^{13}\text{C}_2$ -Asp are fully consistent with our conclusion of Asp production from $^{13}\text{C}_5$ -Glu through the TCA cycle.

Second, and in line with the Reviewer's comment, in the revised manuscript we have now investigated various other pathways that produce Asp from Glu (new Fig. S8). Thus, a TCA cycle-independent pathway to produce Asp would be through a reductive carboxylation, which produces $^{13}\text{C}_3$ -Asp from $^{13}\text{C}_5$ -Glu. However, as we now show in the new Fig. S8, the amount of $^{13}\text{C}_3$ -Asp is very small when compared to $^{13}\text{C}_2$ -Asp or $^{13}\text{C}_4$ -Asp, and there is no significant difference in $^{13}\text{C}_3$ -Asp between WT and ARHGAP11B+GLUD2 bRG. In addition, we have examined a possible contribution of the GABA shunt pathway, another pathway transferring carbon atoms of Glu to Asp through a segment of the TCA cycle. However, neither $^{13}\text{C}_4$ -GABA nor $^{13}\text{C}_2$ -GABA showed significant changes between WT and ARHGAP11B+GLUD2 bRG (new Fig. S8). In addition, the amount of $^{13}\text{C}_4$ -GABA and $^{13}\text{C}_2$ -GABA were really small compared to $^{13}\text{C}_2$ -Asp or $^{13}\text{C}_4$ -Asp. Therefore, we think that it is fair to conclude that the major active pathway in the ARHGAP11B+GLUD2 bRG to produce Asp is the one through the TCA cycle (both full TCA cycle followed by a $\frac{3}{4}$ TCA cycle, and just a $\frac{3}{4}$ TCA cycle). However, since we feel that it is beyond the scope of the present study to test all possible minor pathways of Asp production, we have softened our conclusion and now state "*ARHGAP11B+GLUD2 bRG seem to exhibit a greater degree of glutamate-to- αKG conversion followed by either one $\frac{3}{4}$ TCA cycle or one full TCA cycle followed by one $\frac{3}{4}$ TCA cycle, to produce more aspartate*" (page 10).

Third, and in line with the Reviewer's comment, in the revised manuscript we have performed additional metabolomic analyses that take normal ^{12}C metabolites into consideration (new Fig. S8c-f).

We sincerely hope that with these explanations and additional data, we have satisfactorily addressed the Reviewer's concerns.

Reviewer's comments:

Additionally, the statement that aspartate is an essential amino acid is incorrect, as essentially all human cells can synthesize aspartate, making it a non-essential amino acid. The study shows that Aps is derived from Glu, suggesting that the cells studied can make it.

Authors' Response:

We apologize for using the word "essential" in a misleading context. The Reviewer is correct, of course, that aspartate is not an essential amino acid. What we meant say is that aspartate is crucial for nucleotide synthesis. We have corrected the text accordingly to avoid any misunderstanding.

Reviewer's comments:

Given the stunning morphological changes, it's unlikely that the minor change in aspartate is a driving factor. It is more plausible that the genetic alteration causes changes in neocortex development that subsequently affect the glutamine to aspartate ratio. In general, small changes in genes and gene expressions that are important for development can have drastic (>2-fold) effects on metabolic flux. Another weakness in the flux analysis is the fact that the aspartate pathway is not quantified, and from the presented data, it's impossible to determine what fraction of glutamate is actually going towards glutamine, proline, or aspartate. Thus, while this reviewer agrees that there might be a slide shift in $^{13}\text{C}_x$ Asp, the data present are insufficient to support the strong statements made.

Authors' Response:

We are not sure what the Reviewer refers to with "*stunning morphological changes*". What we demonstrate is an increased proliferation of bRG upon ARHGAP11B and GLUD2 expression, which then leads to a greater abundance of bRG in the ARHGAP11B+GLUD2 transgenic embryos.

We agree with the Reviewer that small changes in gene expressions can have drastic effects on metabolic flux, which then could lead to increased progenitor abundance. In fact, we believe that it is the expression of ARHGAP11B and GLUD2 in the double-transgenic mice which underlies the observed metabolic effects that eventually result in increased bRG abundance.

To address the Reviewer's criticism regarding metabolite quantification, in the revised manuscript the data in revised Fig. 3 and the new Fig. S8 are now presented as ratios to $^{13}\text{C}_5$ -Glu, in order to show the change in metabolic flux in the transgenic mice compared to the wildtype mice. In addition, we have now added the original data showing the absolute abundance of analyzed metabolites in the new Fig. S8, panels e and f. Furthermore, we provide additional data on the fraction of glutamate that is converted to the various metabolites analyzed in ARHGAP11B+GLUD2 vs. wildtype mouse embryos.

To further address the Reviewer's comment, we have performed the following experiment to test if aspartate is required for the proliferation of bRG. Fetal human neocortical tissues *ex vivo* were treated with a selective inhibitor to reduce the level of aspartate production from oxaloacetate. We observed that the abundance of bRG was significantly reduced in fetal human neocortex *ex vivo* (new panels g-i of revised Fig. 3). These additional data further support our conclusion that aspartate production from oxaloacetate contributes to increasing bRG abundance, and demonstrate that this metabolic pathway is relevant for the developing human neocortex.

Reviewer's comments:

In the opinion of this reviewer, the statement that the genetic alteration induces glutamate to aspartate flux via the TCA cycle is not sufficiently supported by the data as presented especially since transaminases and other common housekeeping enzymes can produce the observed flux.

Authors' Response:

We have addressed this comment of the Reviewer already above. Specifically, as described above, we have interrogated the possible contributions of the TCA cycle, the reductive carboxylation pathway, and the GABA shunt pathway as the carbon source of Asp. Of those pathways, the carbon flux data clearly showed that the Asp production from Glu through the TCA cycle is the main pathway promoted upon ARHGAP11B and GLUD2 expression.

As the Reviewer correctly pointed out, glutamic-oxaloacetic transaminases (GOT1/2) are the critical enzymes

in Asp production from oxaloacetate (OAA). To examine the contribution of GOT1/2 in the Asp production in bRG, fetal human neocortical tissues were treated with a selective inhibitor of GOT1/2. We observed that the abundance of bRG was significantly reduced in fetal human neocortical tissue *ex vivo* upon treatment with this inhibitor, indicating that GOT1/2 play an important role in the Asp production in bRG (revised Fig. 3, new panels g-i). Importantly, if Asp production from OAA using Glu were to occur without the OAA first having been generated from $^{13}\text{C}5\text{-Glu}$ via αKG and a 3/4 TCA or a full TCA cycle followed by a 3/4 TCA cycle, then the Asp would not contain $^{13}\text{C}4$ or $^{13}\text{C}2$, respectively, which however we found to be the case. In other words, in this metabolic pathway, all carbon atoms of Asp originate from oxaloacetate, whereas GOT1/2 converts oxaloacetate to Asp by transferring an amino group from Glu to Asp. Hence, our data using $^{13}\text{C}5\text{-Glu}$ indicate that the TCA cycle was involved in the generation of OAA which was then used to produce Asp. We conclude that the contribution of GOT1/2 is fully consistent with our conclusion that Asp production from Glu occurs via αKG and the TCA cycle.

Reviewer #3

Reviewer's comments:

In this original work, to help identify the mechanism of action of ARHGAP11B, authors focused on primate-specific metabolic regulators in mitochondria in neural progenitor cells. GLUD2 was identified as acting synergistically to ARHGAP11B. Extracellular glutamate is also required to increase bRG abundance (including uptake via GLAST).

The authors performed transgenic expression in mouse neocortex (mouse lines as well as IUE). The combination of GLUD2 and ARHGAP11B leads to an increased production of bRGs. The Glast KO reduced bRG production, and in human fetal brain tissue a glutamate uptake inhibitor also specifically reduced bRG. RNAseq data (previously published) suggests bRGs have high glutamate influx and neurons high efflux. Metabolic pathways were investigated downstream of glutamate in bRG in the ARHGAP11B and GLUD2 double-transgenic mouse embryos. Significantly increased aspartate levels (and decreased proline levels) were identified in ARHGAP11B+GLUD2 bRG compared to WT bRG (and higher in bRG than aRG) suggesting a greater degree of glutamate-to-aKG conversion. aKG supplementation was then found to be sufficient to increase the abundance of BPs, including that of bRG in mouse cortex. aKG supplementation also rescued the phenotype caused by a disruption of ARHGAP11B function in fetal human neocortex.

The results are interesting describing a functional synergy between two proteins in the evolutionary expansion of the human neocortex and changes in metabolism, which have arisen during evolution.

Authors' Response:

We greatly appreciate that the Reviewer finds our results "interesting", and thank the Reviewer for this concise summary of our study.

Reviewer's comments:

Comments and questions

Does ARHGAP11B interact physically with GLUD2 (and /or other enzymes in the glutaminolysis or TCA pathway)? Does this happen only in the mitochondria or to some extent also in the cytosol? Are there other proteins (not necessarily primate-specific) which may positively influence the function of ARHGAP11B in these pathways?

Authors' Response:

The Reviewer raises interesting questions, which we would like to address as follows. In our previous study (Namba et al., *Neuron* 105:867-881, 2020), we performed an ARHGAP11B pulldown analysis to identify all binding partners of ARHGAP11B. We did not identify any glutaminolysis-related proteins, including glutamate dehydrogenase (GLUD1/2) and glutamic-oxaloacetic transaminases (GOT1/2), with the adenine nucleotide translocase being the major binding partner. Therefore, ARHGAP11B is not likely to physically interact with GLUD2.

Both ARHGAP11B and GLUD2 are destined to be located in mitochondria, which implies that the effects of ARHGAP11B and GLUD2 on metabolism happen primarily in the mitochondria.

Since inhibition of GOT1/2, which are evolutionarily conserved enzymes catalyzing aspartate production from oxaloacetate, in fetal human neocortical tissues *ex vivo* resulted in a reduction in bRG abundance (revised Fig. 3, new panels g-i), the pathway upregulated by ARHGAP11B and GLUD2 utilizes GOT1/2 for the aspartate production.

Reviewer's comments:

GLAST is mentioned but might other SLCs also be expected to be important to enhance production or

maintenance of bRGs?

Authors' Response:

The Reviewer, again, raises an important question. To this end, we have analyzed two publicly available RNA sequencing datasets for the expression of *SLC1A1*, *SLC1A2* (encoding GLT-1) and *SLC1A3* (GLAST). *SLC1A1* is not expressed in the developing human neocortex to any significant extent. Although *SLC1A2* is also expressed in human bRG, *SLC1A3* is expressed more highly than *SLC1A2* in bRG (new Fig. S7, a and b). As for the protein expression of GLT-1, previous studies showed that GLT-1 is transiently and mainly expressed in neurons during the neurogenic period in mice, rat and sheep (Furuta et al., *J. Neurosci.* 17:8367-8375, 1997; Northington et al., *J. Neurobiol.* 39:515-526, 1999; Matsugami et al., *PNAS* 103:12161-12166, 2006). We therefore conclude that GLAST is the dominant glutamate transporter expressed in bRG.

Reviewer's comments:

Figure 1 f,g,h and page 7: What about overall numbers of Pax6 + cells? It is important to show this data and a binning analysis. Do double transgenics have a thinner VZ?

Authors' Response:

In response to the Reviewer's fair questions, we would like to provide the following information. First, the total number of Pax6+Tbr2- cells in the VZ and SVZ was not changed among the four genotypes (new Fig. S3a). Second, a binning analysis of the Pax6+Tbr2- cells showed that there is a small, but significant increase in the number of the Pax6+Tbr2- cells in the basal part of the SVZ (new Fig. S3b). Third, the thickness of the VZ was not changed among the four genotypes (new Fig. S3c).

Reviewer's comments:

ARHGAP11B / GLUD2 immunohistochemistry – are these two proteins only expressed in a proportion of bRGs (c.f. Sox2 labelling Supp Fig 1)? Signals in more apical regions seem quite large and diffuse, as well as more intense, why is this? Also, how is the distribution in soma versus basal processes – it would be good to see co-staining with a mitochondrial marker.

Authors' Response:

Again, in response to the Reviewer's fair questions, we would like to provide the following information. First, we have performed triple immunofluorescence with antibodies recognizing SOX2, TOM20 (a mitochondrial marker), and ARHGAP11B or GLUD2. We show a colocalization of ARHGAP11B or GLUD2 with Tom20 in the SOX2+ OSVZ cells (new Fig. S1j-l). As the Reviewer correctly pointed out, the subcellular distribution of ARHGAP11B and GLUD2 in bRG (soma vs. processes) is could be relevant. However, since (for reasons of the species used to generate the antibodies) SOX2 (as a nuclear protein) is the only bRG marker we can use for the triple immunofluorescence, analyzing the localization of ARHGAP11B and GLUD2 in the process of bRG is difficult. Nevertheless, we found that both ARHGAP11B and GLUD2 are partly localized in the proximal part of the basal process (revised Fig. S1, new panels j and k).

Second, regarding the proportion of bRG concerned, we have quantified the percentage of ARHGAP11B+ and/or GLUD2+ cells among Sox2+ cells in the OSVZ. Approximately 80-90% of Sox2+ cells were found to be positive for ARHGAP11B and GLUD2 (new Fig. S11).

Third, regarding the signal in the apical region, we indeed do find stronger/more intense signals along the apical surface. We believe there is a difference in the expression/protein level of ARHGAP11B and GLUD2 among cells in different cell cycle phases (higher expression in progenitors in M-phase, which are located at the apical surface during mitosis). Another explanation could be that the dividing cells have more densely packed mitochondria, which could contribute to the more intense apical signal of *ARHGAP11B / GLUD2* immunofluorescence.

Reviewer's comments:

Is there an impact of transgenic expression on cells in mouse brain (transgenes expressed in APs as well as BPs), do cells change in size?, is there an impact on number and morphology of mitochondria?

Authors' Response:

Within the densely packed neocortical tissue, we find it difficult to evaluate cell size directly.

We have examined mitochondrial density in the VZ and SVZ by TOM20 immunofluorescence. There were no significant changes in the fluorescent intensity of TOM20 (new Fig. S4a and b), suggesting that the number of mitochondria is not affected. In addition, we have analyzed the morphology of mitochondria in the VZ and SVZ, and found that the length of mitochondria is not changed in the *ARHGAP11B+GLUD2* double-transgenic mouse embryos (new Fig. S4c). Taken together, these data suggest that the number and morphology of mitochondria are not changed in the *ARHGAP11B+GLUD2* double-transgenic embryos.

Reviewer's comments:

Page 8, conclusion: Taken together, our results suggest that in fetal human neocortex, bRG take up extracellular glutamate, ... through GLAST to maintain their abundance, which largely reflects the action of ARHGAP11B plus the enhancing effect of GLUD2'. There is only indirect evidence to suggest that glutamate uptake is required to maintain bRG abundance - it is not confirmed that there is an effect on proliferation, and what about the possibility that there is an impact on bRG production from aRG? It is interesting that aRGs in the double transgenic model also show metabolic changes compared to WT in the double transgenic (Supp Fig 5b), even if it is less /different than the effect on bRGs. This impact on abundance (compared to production) needs to be shown more conclusively.

Authors' Response:

We respectfully disagree with the Reviewer's comment that "*There is only indirect evidence to suggest that glutamate uptake is required to maintain bRG abundance*". The Glutamate Aspartate Transporter (GLAST) is the main transporter for bRG to take up glutamate in the developing neocortex. When we knock out GLAST by crossing the *ARHGAP11B+GLUD2* double-transgenic mice with GLAST KO mice, we did not observe the increase in bRG abundance induced by the synergy of *ARHGAP11B+GLUD2* anymore, without any effect of the GLAST KO on apical progenitors and basal intermediate progenitors. Hence, this data is one line of direct evidence indicating that glutamate uptake is required to maintain bRG abundance (Fig. 2c). Moreover, we used a GLAST inhibitor (PDC) to block GLAST function in human neocortical tissue *ex vivo*, which also reduced the abundance of bRG (Fig. 2g and h). This therefore is an independent line of evidence indicating that glutamate uptake is required to maintain bRG abundance.

To address the Reviewer's suggestion that the increase in bRG abundance may be due to increased generation from aRG, we examined whether the delamination of aRG, which leads to the production of bRG, is increased upon *ARHGAP11B* and *GLUD2* expression, using γ -tubulin immunostaining as previously published (Tavano et al., *Neuron* 97:1299-1314, 2018). However, we did not observe significant changes in the number of abventricular γ -tubulin in the *ARHGAP11B+GLUD2* double-transgenic neocortex (new Fig. S3d and e), suggesting that an increased proliferation of bRG, rather than a production of bRG from aRG by delamination, is the main cause of the increased abundance of bRG in the *ARHGAP11B+GLUD2* double-transgenic neocortex.

Reviewer's comments:

Even if aspartate is stated to be essential for proliferation, increased proliferation of bRGs needs to be shown here to support the conclusions.

Authors' Response:

Using immunofluorescence for the mitotic marker phospho-vimentin, we have shown that the number of mitotic

bRG is significantly increased in the *ARHGAP11B+GLUD2* double-transgenic mouse embryos when compared to WT or to *ARHGAP11B* and *GLUD2* single-transgenic embryos (Fig. 1b, e). These data indicate that the proliferation of bRG is increased upon ARHGAP11B and GLUD2 expression.

Furthermore, in line with the Reviewer's suggestion, we have performed the following experiment to test if aspartate is required for the proliferation of bRG. Fetal human neocortical tissues were treated with a selective inhibitor *ex vivo* to reduce the level of aspartate production. We observed that the abundance of bRG was significantly reduced in the fetal human neocortex upon the inhibitor treatment (revised Fig. 3, new panels g-i). These data suggest that aspartate is required for the proliferation of bRG in the fetal human neocortex.

Reviewer's comments:

In addition, as shown in Fig 4a-d, bIPs as well as bRGs are increased with added α KG supplementation and there is also a tendency for increase in aRGs. Does the double transgenic therefore increase BP production?

Authors' Response:

The results in Fig. 1d and e indicate that whereas the abundance of bRG in the ARHGAP11B+GLUD2 double-transgenic embryos was increased when compared to all other genotypes (Fig. 1e), the abundance of bIPs was not significantly different between ARHGAP11B+GLUD2 double-transgenic embryos and ARHGAP11B single-transgenic embryos. These results suggest that the effect on bIP abundance is mainly due to the expression of ARHGAP11B alone.

In the experiments shown in Fig. 4a-d, we used ETaKG, which enters all cells, including bIPs, and could stimulate the proliferation of any type of cycling cells. We find it likely that this general entry of ETaKG into cells is the reason why we observed an increase in bIP abundance (Fig. 4c) in addition to bRG abundance (Fig. 4d).

We mainly focused our study on bRG because bRG are the mouse evolutionarily relevant neural progenitor cell type for neocortex development and the key determining cell population for neocortex expansion during evolution, especially between mouse and human.

Reviewer's comments:

Minor

In several experiments, bRGs or aRGs are purified for the analyses. It would be important to show data convincing the reader of the quality of this purification.

Authors' Response:

We have followed previously published protocols for isolating bRG and aRG from embryonic mouse neocortex (Pinson et al., *Science* 377: eabl6422, 2022). In the revised manuscript, we have now included the FACS sorting plots in the supplement to illustrate our sorting strategies (gating, negative control and sorting efficiency; please see new Fig. S5) to convince the Reviewer (and hopefully the readers) of the quality of our progenitor cell isolation.

Reviewer's comments:

Authors say on page 6, transgenes are similar to the expression pattern observed in fetal human neocortex (Fig. S1c-i) – but they don't show this.

Authors' Response:

We have added immunofluorescence data on the expression ARHGAP11B and GLUD2 in the double-transgenic mouse neocortex (revised Fig. S1, new panel m). The ARHGAP11B and GLUD2 immunofluorescence signals were observed in the Sox2+ cells located in the VZ and SVZ. The expression

pattern was similar to what we observed in fetal human neocortex.

Reviewer's comments:

GLUD2 sounds like glutamate receptor delta, it would be important to incorporate the full name in the abstract.

Authors' Response:

We thank the Reviewer for this comment. We now mention the full name of GLUD2 in the abstract to avoid potential confusion.

Reviewer's comments:

Fig 1: it is written that 'the number of pVim+ abventricular cells without basal and/or apical processes, considered to be mitotic bIPs, ... the number of pVim+ abventricular cells with basal and/or apical processes, considered to be mitotic bRG', these two categories overlap? Need to be clearer.

Authors' Response:

These two categories do not overlap, as they are clearly distinguished from each other by the absence vs. presence of basal and or apical processes during mitosis. The presence of basal processes during mitosis is a key feature of bRG, compared to bIPs, as reported previously (Wang et al, *Nat. Neurosci.* 14:555-561 2011).

Reviewer's comments:

Fig 2: bRG can uptake glutamate, but we are only told in Supp Fig 4 (RNA seq analysis), how this compares to bIPs, I would suggest to move this information up in the manuscript, otherwise testing glutamate uptake in bIPs and aRGs using the same methods as for bRGs, could become a necessity. aRGs are not mentioned in the text, although this information is incorporated in Supp Fig 4. It would be good to mention aRGs as well.

Authors' Response:

We have shown glutamate can be taken by bRG in Fig. 2a, as labeled ¹³C5-Glu was detected in both WT and double-transgenic bRG. In addition, we have cited the new Fig. S7 and mentioned this information earlier in the manuscript. In line with the Reviewer's suggestion, we now mention aRG in the text as well.

Reviewer's comments:

How does Glast expression compare in the cell types? The SLC which is GLAST should be indicated clearly in the legend.

Authors' Response:

GLAST is expressed in both aRG and bRG, but not in bIPs and neurons in the developing neocortex (Matsugami et al., *PNAS* 103:12161-12166, 2006; please see also new Fig. S6a and b). In line with the Reviewer's suggestion, we have now clearly indicated that GLAST is *SLC1A3* in the figure legend.

Reviewer's comments:

Fig 4e-h, why were PCNA+ cells analysed specifically?

Authors' Response:

PCNA is a cycling cell marker protein, which is expressed in almost all proliferating cells in the developing neocortex (Arai et al., *Nat. Commun.* 2:154, 2011). We used PCNA in Fig 4e-h to distinguish the cycling progenitors from neurons (which are PCNA negative) in the SVZ.

Reviewer's comments:

Was the sex of the mouse embryos assessed or of the human fetal cortex known?

Authors' Response:

We did not assess the sex of mouse embryos used for experiments as we believe there is no sex difference in bRG abundance at this early stage during development when we performed our analysis. The sex of human fetal cortex could not be assessed because the informed consent form used in Finland did not indicate that the tissue would be used for chromosomal analyses. In this context, we would like to emphasize that mouse embryos and fetal human neocortical tissues were collected randomly. So any sex difference pertaining to our data would be included in the standard variation of our data.

Reviewer's comments:

The English is good, but another read through by a native English speaker could be good

Authors' Response:

We have asked our native English speaker colleagues to proofread our revised manuscript and have incorporated their comments.

REVIEWERS' COMMENTS

Reviewer #1 (Remarks to the Author):

I reread the manuscript and looked at their response to the comments. I think they answered all of my comments well, and addressed them thoroughly.

Reviewer #2 (Remarks to the Author):

The author has addressed all concerns raised by the reviewer to full satisfaction, and it is recommended for application.